# Identifiability in inverse reinforcement learning

**Haoyang Cao**
Alan Turing Institute
hcao@turing.ac.uk

**Samuel N. Cohen**
Mathematical Institute, University of Oxford and Alan Turing Institute
samuel.cohen@maths.ox.ac.uk

**Łukasz Szpruch**
School of Mathematics, University of Edinburgh and Alan Turing Institute
L.Szpruch@ed.ac.uk

## Abstract

Inverse reinforcement learning attempts to reconstruct the reward function in a Markov decision problem, using observations of agent actions. As already observed in Russell [1998] the problem is ill-posed, and the reward function is not identifiable, even under the presence of perfect information about optimal behavior. We provide a resolution to this non-identifiability for problems with entropy regularization. For a given environment, we fully characterize the reward functions leading to a given policy and demonstrate that, given demonstrations of actions for the same reward under two distinct discount factors, or under sufficiently different environments, the unobserved reward can be recovered up to a constant. We also give general necessary and sufficient conditions for reconstruction of time-homogeneous rewards on finite horizons, and for action-independent rewards, generalizing recent results of Kim et al. [2021] and Fu et al. [2018].

## 1 Introduction

Inverse reinforcement learning aims to use observations of agents' actions to determine their reward function. The problem has roots in the very early stages of optimal control theory; Kalman [1964] raised the question of whether, by observation of optimal policies, one can recover coefficients of a quadratic cost function (see also Boyd et al. [1994]). This question naturally generalizes to the generic framework of Markov decision process and stochastic control.

In the 1970s, these questions were taken up within economics, as a way of determining utility functions from observations. For instance, Keeney and Raiffa [1976] set out to determine a proper ordering of all possible states which are deterministic functions of actions. In this setup, the problem is static and the outcome of an action is immediate. Later in Sargent [1978], a dynamic version of a utility assessment problem was studied, under the context of finding the proper wage through observing dynamic labor demand.

As exemplified by Lucas' critique[1], in many applications it is not enough to find *some* pattern of rewards corresponding to observed policies; instead we may need to identify *the specific* rewards agents face, as it is only with this information that we can make valid predictions for their actions in a changed environment. In other words, we do not simply wish to learn a reward which allows us to imitate agents in the current environment, but which allows us to predict their actions in other settings.

---

[1]The critique is best summarized by the quotation: "Given that the structure of an econometric model consists of optimal decision rules of economic agents, and that optimal decision rules vary systematically with changes in the structure of series relevant to the decision maker, it follows that any change in [regulatory] policy will systematically alter the structure of econometric models." (Lucas [1976])

In this paper, we give a precise characterization of the range of rewards which yield a particular policy for an entropy regularized Markov decision problem. This separates the main task of estimation (of the optimal policy from observed actions) from the inverse problem (of inferring rewards from a given policy). We find that even with perfect knowledge of the optimal policy, the corresponding rewards are not fully identifiable; nevertheless, the space of consistent rewards is parameterized by the value function of the control problem. In other words, the reward can be fully determined given the optimal policy and the value function, but the optimal policy gives us no direct information about the value function.

We further show that, given knowledge of the optimal policy under two different discount rates, or sufficiently different transition laws, we can uniquely identify the rewards (up to a constant shift). We also give conditions under which action-independent rewards, or time-homogenous rewards over finite horizons, can be identified. This demonstrates the fundamental challenge of inverse reinforcement learning, which is to disentangle immediate rewards from future rewards (as captured through preferences over future states).

## 2 Background on reinforcement learning

The motivation behind inverse reinforcement learning is to use observed agent behavior to identify the rewards motivating agents. Given these rewards, one can forecast future behavior, possibly under a different environment. In a typical reinforcement learning[2] (RL) problem, an agent learns an optimal policy to maximize her total reward by interacting with the environment.

In order to analyse the inverse reinforcement learning problem, we begin with an overview of the 'primal' problem, that is, how to determine optimal policies given rewards. We particularly highlight a entropy regularized version of the Markov decision process (MDP), which provides a better-posed setting for inverse reinforcement learning. For mathematical simplicity, we focus on discrete-time problems with finitely many states and actions; our results can largely be transferred to continuous settings, with fundamentally the same proofs, however some technical care is needed.

### 2.1 Discrete Markov decision processes with entropy regularization

**The environment.** We consider a simple Markov decision process (MDP) on an infinite horizon. The MDP $\mathcal{M} = (\mathcal{S}, \mathcal{A}, \mathcal{T}, f, \gamma)$ is described by: a finite state space $\mathcal{S}$; a finite set of actions $\mathcal{A}$; a (Markov) transition kernel $\mathcal{T} : \mathcal{S} \times \mathcal{A} \to \mathcal{P}(\mathcal{S})$, that is, a function $\mathcal{T}$ such that $\mathcal{T}(s, a)$ gives probabilities[3] of each value of $S_{t+1}$, given the state $S_t = s$ and action $A_t = a$ at time $t$; and a reward function $f : \mathcal{S} \times \mathcal{A} \to \mathbb{R}$ with discount factor $\gamma \in [0, 1)$.

An agent aims to choose a sequence of actions $\{A_0, A_1, ...\}$ from $\mathcal{A}$ in order to to maximize the expected value of total reward

$$\sum_{t=0}^{\infty} \gamma^t f(S_t, A_t).$$

It will prove convenient for us to allow randomized policies $\pi$, that is, functions $\pi : \mathcal{S} \to \mathcal{P}(\mathcal{A})$, where $\pi(\cdot|s)$ is the distribution of actions the agent takes when in state $s$. For a given randomized policy $\pi$, we define $\mathcal{T}_\pi \in \mathcal{P}(\mathcal{S})$, the distribution of $S_{t+1}$ given state $S_t$, by

$$\mathcal{T}_\pi(S_{t+1} = s'|S_t = s) = \sum_{a \in \mathcal{A}} \mathcal{T}(s'|s, a)\pi(a|s).$$

Given an initial distribution $\rho \in \mathcal{P}(\mathcal{S})$ for $S_0$ and a policy $\pi : \mathcal{S} \to \mathcal{P}(\mathcal{A})$, we obtain a (unique) probability measure $\mathbb{P}_\rho^\pi$ such that, and any $a \in \mathcal{A}, s, s' \in \mathcal{S}$, $\mathbb{P}_\rho^\pi(S_0 = s) = \rho(s)$ and

$$\mathbb{P}_\rho^\pi(A_t = a|S_t = s) = \pi(a|s), \quad \mathbb{P}_\rho^\pi(S_{t+1} = s'|S_t = s, A_t = a) = \mathcal{T}(s'|s, a), \text{ for all } t.$$

We write $\mathbb{E}_\rho^\pi$ for the corresponding expectation and $\mathbb{E}_s^\pi$ when the initial state is given by $s \in \mathcal{S}$. The classic objective in a MDP is to maximize the expected value $\mathbb{E}_s^\pi \left[ \sum_{t=0}^{\infty} \gamma^t f(S_t, A_t) \right]$. With this

---

[2]Reinforcement learning and optimal control are closely related problems, where reinforcement learning typically focuses on the challenge of numerically learning a good control policy, while optimal control focuses on the description of the optimizer. In the context of the inverse problem we consider they are effectively equivalent and we will use the terms interchangeably.

[3]Here, and elsewhere, we write $\mathcal{P}(X)$ for the set of all probability distributions on a set $X$.

objective, one can show (for example, see Bertsekas and Shreve [2004], Puterman [2014]) that there is an optimal deterministic control (i.e. a policy $\pi$, taking values zero and one, which maximizes the expected value). This implies that, typically, an optimal agent will only make use of a single action for each state, and the choice of this action will not vary smoothly with changes in the reward, discount rate, or transition kernel.

**Entropy regularised MDP.** Given the lack of smoothness in the classical MDP, and to encourage exploration, a well-known variation on the classic MDP introduces a regularization term based on the Shannon entropy. Given a policy $\pi$ and regularization coefficient $\lambda \geq 0$, the entropy regularized value of a policy $\pi$, when starting in state $s$, is defined by

$$V_\lambda^\pi(s) := \mathbb{E}_s^\pi\left[\sum_{t=0}^\infty \gamma^t\left(f(s_t, a_t) - \lambda \log\left(\pi(a_t|s_t)\right)\right)\right] = \mathbb{E}_s^\pi\left[\sum_{t=0}^\infty \gamma^t\left(f(s_t, a_t) + \lambda \mathcal{H}\left(\pi(\cdot|s_t)\right)\right)\right].$$
(1)

Here $\mathcal{H}(\pi) = -\sum_{a \in \mathcal{A}} \pi(a) \log(\pi(a))$ is the entropy of $\pi$. We call this setting the regularised MDP $\mathcal{M}_\lambda = (\mathcal{S}, \mathcal{A}, \mathcal{T}, f, \gamma, \lambda)$. The optimal value is given by $V_\lambda^*(s) := \max_\pi V_\lambda^\pi(s)$, where the maximum is taken over all (randomized feedback[4]) policies $\pi : \mathcal{S} \to \mathcal{P}(\mathcal{A})$.

We define the state-action value of $\pi$ at $(s, a) \in \mathcal{S} \times \mathcal{A}$ by

$$Q_\lambda^\pi(s, a) = f(s, a) + \gamma \sum_{s' \in \mathcal{S}} \mathcal{T}(s'|s, a) V_\lambda^\pi(s').$$
(2)

The dynamic programming principle (e.g. Haarnoja et al. [2017, Theorem 2]) gives

$$V_\lambda^*(s) = \max_{m \in \mathcal{P}(\mathcal{A})}\left[\sum_{a \in \mathcal{A}}\left(f(s, a) - \lambda \log\left(m(a)\right) + \gamma \mathbb{E}_{s_1 \sim \mathcal{T}(\cdot|s,a)}\left[V_\lambda^*(s_1)\right]\right) m(a)\right]$$

$$= \lambda \max_{m \in \mathcal{P}(\mathcal{A})}\left[\frac{1}{\lambda}\sum_{a \in \mathcal{A}}\left(f(s, a) + \gamma \mathbb{E}_{s_1 \sim \mathcal{T}(\cdot|s,a)}\left[V_\lambda^*(s_1)\right]\right) m(a) + \mathcal{H}(m)\right].$$
(3)

Observing that on the right hand side we are maximizing over a linear function in $m$ plus an entropy term, and applying [Dupuis and Ellis, 2011, Proposition 1.4.2], we have that for any $s \in S$

$$V_\lambda^*(s) = V_\lambda^{\pi_\lambda^*}(s) = \lambda \log \sum_{a \in \mathcal{A}} e^{\frac{1}{\lambda}\left(f(s,a) + \gamma \mathbb{E}_{s_1 \sim \mathcal{T}(\cdot|s,a)}[V_\lambda^{\pi_\lambda^*}(s_1)]\right)} = \lambda \log \sum_{a \in \mathcal{A}} \exp\left(\frac{1}{\lambda} Q_\lambda^{\pi_\lambda^*}(s, a)\right),$$
(4)

and the maximum in (3) is achieved by the randomized policy $m(a) = \pi_\lambda^*(a|s)$, where

$$\pi_\lambda^*(a|s) = \frac{\exp\left(\frac{1}{\lambda} Q_\lambda^{\pi_\lambda^*}(s, a)\right)}{\sum_{a' \in \mathcal{A}} \exp\left(\frac{1}{\lambda} Q_\lambda^{\pi_\lambda^*}(s, a')\right)} \qquad \text{for } a \in \mathcal{A}.$$
(5)

From (4) we see that $\exp\left(V_\lambda^*(s)/\lambda\right) = \sum_{a \in \mathcal{A}} \exp\left(Q_\lambda^{\pi_\lambda^*}(s, a)/\lambda\right)$ and so we can write the optimal policy as

$$\pi_\lambda^*(a|s) = \exp\left(\left(Q_\lambda^{\pi_\lambda^*}(s, a) - V_\lambda^*(s)\right)/\lambda\right)$$

$$= \exp\left(\left(f(s, a) + \mathbb{E}_{s_1 \sim \mathcal{T}(\cdot|s,a)}[\gamma V_\lambda^*(s_1) - V_\lambda^*(s)]\right)/\lambda\right),$$
(6)

From this analysis, we make the following observations regarding the regularized MDP:

- The optimal policy will select all actions in $\mathcal{A}$ with some positive probabilities.
- If $\lambda$ is increased, this has the effect of 'flattening out' the choice of actions, as seen in the softmax function in (5). Conversely, sending $\lambda \to 0$ will result in a true maximizer being chosen, and the regularized problem degenerates to the classical MDP.
- Adding a constant to the reward does not change the policy.

**Remark 1.** *In many modern approaches, one replaces dependence on the state with dependence on a space of 'features'. This has benefits when fitting a model, but does not significantly change the problem considered.*

---

[4]Given the Markov structure there is no loss of generality when restricting to policies of feedback form. Further, by replacing $\mathcal{A}$ with the set of maps $\mathcal{S} \to \mathcal{A}$ if necessary, all feedback controls $a(s)$ can be written as deterministic controls $a(\cdot)$ in a larger space, so when convenient we can consider controls which do not depend on the state without loss of generality.

# 3 Analysis of inverse reinforcement learning

We now shift our focus to 'inverse' reinforcement learning, that is, the problem of inferring the reward function given observation of agents' actions.

Consider a discrete time, finite-state and finite-action MDP $\mathcal{M}_\lambda$, as described in Section 2. Suppose a 'demonstrator' agent acts optimally, and hence generates a *trajectory* of states and actions for the system $\tau = (s_1, a_1, s_2, a_2, ...)$. We assume that it is possible for us to observe $\tau$ (over a long period), and seek to infer the reward $f$ which the agent faces.

A first observation is that, assuming each state $s \in \mathcal{S}$ appears infinitely often in the sequence $\tau$, and the agent uses a randomized feedback control $\pi_\lambda(a|s)$, it is possible to infer this control. A simple consistent estimator for the control is

$$(\pi_\lambda)_N(a|s) = \frac{\#\{a_t = a \text{ and } s_t = s; \quad t \leq N\}}{\#\{s_t = s; \quad t \leq N\}} \to \pi_\lambda(a|s) \text{ a.s. as } N \to \infty.$$

Similarly, assuming each state-action pair $(s, a)$ appears infinitely often in $\tau$, we can infer the controlled transition probabilities $\mathcal{T}(s'|s, a)$. A simple consistent estimator is given by

$$\mathcal{T}_N(s'|s, a) = \frac{\#\{s_t = s, a_t = a \text{ and } s_{t+1} = s'; \quad t \leq N\}}{\#\{s_t = s \text{ and } a_t = a; \quad t \leq N\}} \to \mathcal{T}(s'|s, a) \text{ a.s. as } N \to \infty.$$

If our agent is known to follow a regularized optimal strategy, as in (6), and we have a simple accessibility condition[5] on the underlying states, then every state-action pair will occur infinitely often in the resulting trajectory. Therefore, given sufficiently long observations, we will know the values of $\pi(a|s)$ and $\mathcal{T}(s'|s, a)$ for all $s, s' \in \mathcal{S}$ and $a \in \mathcal{A}$.

This leads, naturally, to an abstract version of the inverse reinforcement learning problem: Given knowledge of $\pi(a|s)$ and $\mathcal{T}(s'|s, a)$ for all $s, s' \in \mathcal{S}$ and $a \in \mathcal{A}$, and assuming $\pi$ is generated by an agent following an entropy-regularized MDP $\mathcal{M}_\lambda$, can we determine the initial reward function $f$ that the agent faces?

As observed by Kalman [1964], for an unregularized controller the only thing we can say is that the observed controls are maximizers of the state-action value function, and not even that these maximizers are unique. Therefore, very little can be said about the underlying reward in the unregularized setting. Indeed, as already observed in Russell [1998] the problem of constructing a reward using state-action data is fundamentally ill-posed. One pathological case is to simply take $f$ constant, so all actions are optimal. Alternatively, if we infer a unique optimal action $a^\star(s)$ for each $s$, we then could take any $f(s, a^\star(s)) \in (0, \infty]$ and $f(s, a) = 0$ for $a \neq a^\star(s)$.

**Further literature.** One of the earliest discussions of inverse reinforcement learning (IRL) in the context of machine learning can be found in Ng and Russell [2000]. Their method is to first identify a class of reward functions, for an IRL problem with finitely many states and actions, a deterministic optimal strategy, and the assumption that the reward function depends only on the state variable. Then, assuming the reward function is expressable in terms of some known basis functions in the state, a linear programming formulation for the IRL problem is presented, to pick the reward function that maximally differentiates optimal policy from the other policies. This characterization of reward functions demonstrates the general non-uniqueness of solutions to IRL problems.

In past two decades, there have been many algorithms proposed to tackle IRL problems. One significant category of algorithms (MaxEntIRL) arises from the maximum entropy approach to optimal control. In Ziebart [2010], IRL problems were linked with maximum causal entropy problems with statistical matching constraints. Similar models can be found in Abbeel and Ng [2004]; Ziebart et al. [2008], Levine et al. [2011] and Boularias et al. [2011]. A connection between maximum entropy IRL and GANs has been established in Finn et al. [2016a]. Further related papers will be discussed in the text below.

---

[5] In particular, for every pair of states $s, s'$, there needs to exist a finite sequence $s = s_1, s_2, ..., s_n = s'$ of states and $a_1, ..., a_{n-1}$ of actions such that $\prod_{k=1}^{n-1} \mathcal{T}(s_{k+1}|s_k, a_k) > 0$. This is certainly the case, for example, if we assume $\mathcal{T}(s'|s, a) > 0$ for all $s, s' \in \mathcal{S}$ and $a \in \mathcal{A}$.

In MaxEntIRL, one assumes that trajectories are generated[6] with a law

$$\mathbb{P}(\tau) = \frac{\rho(s_0)}{Z} \prod_t \mathcal{T}(s_{t+1}|s_t, a_t) e^{\sum_t f(a_t, s_t)}$$

for a constant $Z > 0$. Comparing with the distribution of trajectories from an optimal regularized agent, this approach implicitly assumes that $\pi(a|s) \propto \exp\{f(a, s)\}$. Comparing with (6), this is analogous to assuming the value function is a constant (from which we can compute $Z$) and $\lambda = 1$. This has a concrete interpretation: that many IRL methods make the tacit assumption that the demonstrator agent is myopic. As we shall see in Theorem 1, for inverse RL the value function can be chosen arbitrarily, demonstrating the consistency of this approach with our entropy-regularized agents. We discuss connections with MaxEntIRL further in Appendix C.

### 3.1 Inverse Markov decision problems

We consider a Markov decision problem as in Section 2. As discussed above, we assume that we have full knowledge of $\mathcal{S}, \mathcal{A}, \mathcal{T}, \gamma$, and of the regularization parameter $\lambda$ and the entropy-regularized optimal control $\pi_\lambda$ in (6), but not the reward function $f$.

Our first theorem characterizes the set of all reward functions $f$ which generate a given control policy.

**Theorem 1.** *For a fixed policy $\bar{\pi}(a|s) > 0$, discount factor $\gamma \in [0, 1)$, and an arbitrary choice of function $v : \mathcal{S} \to \mathbb{R}$, there is a unique corresponding reward function*

$$f(s, a) = \lambda \log \bar{\pi}(a|s) - \gamma \sum_{s' \in \mathcal{S}} \mathcal{T}(s'|s, a)v(s') + v(s)$$

*such that the MDP with reward $f$ yields a value function $V_\lambda^{\pi_\lambda^*} = v$ and entropy-regularized optimal policy $\pi_\lambda^* = \bar{\pi}$.*

As a consequence of this theorem, we observe that the value function is not determined by the observed optimal policy, but can be chosen arbitrarily. We also see that the space of reward functions $f$ consistent with a given policy can be parameterized by the set of value functions.

**Remark 2.** *A simple degrees-of-freedom argument gives this result intuitively. There are $n = |\mathcal{S}|$ possible states and $k = |\mathcal{A}|$ possible actions in each state, so the reward function can be described by a vector in $\mathbb{R}^{n \times k}$. From the policy, which satisfies $\sum_{a \in \mathcal{A}} \pi(a|s) = 1$ for all $s$, we observe $n \times (k-1)$ linearly independent values. Therefore, the space of consistent rewards has $n \times k - n \times (k-1) = n$ free variables, which we identify with the $n$ values $\{v(s)\}_{s \in \mathcal{S}}$.*

**Remark 3.** *Ng et al. [1999] provides a useful insight to our result. In Ng et al. [1999] it is assumed that the rewards are of the form $F(S_t, A_t, S_{t+1})$; for a fixed MDP, this adds no generality, as we can write $f(s, a) = \mathbb{E}[F(s, a, S_{t+1})|S_t = s, A_t = a]$. Ng et al. [1999] show that, for any 'shaping potential' $\Upsilon : \mathcal{S} \to \mathbb{R}$, the reward $\tilde{F} = F + \gamma\Upsilon(S_{t+1}) - \Upsilon(S_t)$ yields the same optimal policies for every (unregularized) MDP. However, shaping potentials do not describe the space of all rewards corresponding to a given policy, for fixed transition dynamics. In our results, we instead parameterize a family of costs $f$ in terms of the value function (Theorem 1), and show these are the only costs which lead to the given optimal policy for a fixed (regularized) MDP. We also note that Magnac and Thesmar [2002] give a similar result to ours, using a different randomization of strategies.*

Given Theorem 1, we see that it is not possible to fully identify the reward faced by a single agent, given only observations of their policy. Fundamentally, the issue is that the state-action value function $Q$ combines both immediate rewards $f$ with preferences $v$ over the future state. If we provide data which allows us to disentangle these two effects, for example by considering agents with different discount rates or transition functions, then the true reward can be determined up to a constant, as shown by our next result. In order to clearly state the result, we give the following definition.

**Definition 1.** *Consider a pair of Markov decision problems on the same state and action spaces, but with respective discount rates $\gamma, \tilde{\gamma}$ and transition probabilities $\mathcal{T}, \tilde{\mathcal{T}}$. We say that this pair is value-distinguishing if, for functions $w, \tilde{w} : \mathcal{S} \to \mathbb{R}$, the statement*

$$w(s) - \gamma \sum_{s' \in \mathcal{S}} \mathcal{T}(s'|s, a)w(s') = \tilde{w}(s) - \tilde{\gamma} \sum_{s' \in \mathcal{S}} \tilde{\mathcal{T}}(s'|s, a)\tilde{w}(s') \text{ for all } a \in \mathcal{A}, s \in \mathcal{S} \quad (7)$$

*implies at least one of $w$ and $\tilde{w}$ is a constant function.*

---

[6]As discussed by Levine [2018], for deterministic problems this simplifies to $P(\tau) \propto \exp(\sum_t f(a_t, s_t))$, which is often taken as a starting point.

In this definition, note that constant functions $w, \tilde{w}$ are always solutions to (7), in particular for $c \in \mathbb{R}$ we can set $w \equiv c$ and $\tilde{w} \equiv (1-\gamma)c/(1-\tilde{\gamma})$. However, this is a system of $|\mathcal{A}| \times |\mathcal{S}|$ equations in $2 \times |\mathcal{S}|$ unknowns, so the definition will hold provided our agents' actions have sufficiently varied impact on the resulting transition probabilities. In a linear-quadratic context, it is always enough to vary the discount rates (see Corollary 5).

**Theorem 2.** *Suppose we observe the policies of two agents solving entropy-regularized MDPs, who face the same reward function, but whose discount rates or transition probabilities vary, such that their MDPs are value-distinguishing. Then the reward function consistent with both agents' actions either does not exist, or is identified up to addition of a constant.*

Given the addition of a constant to $f$ does not affect the resulting policy (it simply increases the value function by a corresponding quantity), we cannot expect to do better than Theorem 2 without direct observation of the agent's rewards or value function in at least one state.

**Remark 4.** *Definition 1 is essentially a statement regarding invertibility of a linear system of equations for $w, \tilde{w}$. This indicates that the stability of rewards obtained using Theorem 2 is principally determined by whether this linear system is well conditioned. This can be measured by the ratio of its largest to second smallest singular values (the second smallest is due to the constant functions always being in the kernel of the system) not being too large. Given the inevitable error arising from statistical estimation of policies and transition functions, a well conditioned system is often a key requirement in practice. A similar observation will also be valid for the uniqueness results in later sections.*

**Remark 5.** *Our results show that it is typically sufficient to observe an MDP under* two *environments (transitions and discount factors) in order to identify the reward. This can be contrasted with Amin and Singh [2016] and Amin et al. [2017] who show that, if the demonstrator is observed in multiple (suitably chosen) environments, the (state-only) reward can be identified up to a scaling and shift (the scaling is natural, given they do not use an entropy regularization). Ratliff et al. [2006] consider a finite number of environments, but explicitly do not attempt to estimate the 'true' underlying reward.*

## 4   Finite horizon results

Over finite horizons, for general costs, similar results hold to those already seen on infinite horizons. An entropy-regularized optimizing agent will use a policy $\pi^* = \{\pi_t^*\}_{t=0}^{T-1}$ which solves the following problem with terminal reward $g$ and (possibly time-dependent) running reward $f$:

$$\max_{\pi} \mathbb{E}_s^{\pi} \left[ \sum_{t=0}^{T-1} \gamma^t \Big( f(t, s_t^{\tau}, a_t^{\tau}) - \lambda \log \pi_t(a_t^{\tau}|s_t^{\tau}) \Big) + \gamma^T g(s_T^{\tau}) \right].$$

For any $\pi = \{\pi_t\}_{t=0}^{T-1}$, $s \in \mathcal{S}$, $a \in \mathcal{A}$, and $t \in \{0, \ldots, T-1\}$ write

$$Q_t^{\pi}(s, a) = f(t, s, a) + \gamma \mathbb{E}_{S' \sim \mathcal{T}(\cdot|s,a)} \big[ V_{t+1}^{\pi}(S') \big],$$

$$V_t^{\pi}(s) = \mathbb{E}_{A \sim \pi_t(\cdot|s)} \big[ Q_t^{\pi}(s, A) - \lambda \log \pi_t(A|s) \big], \qquad V_T^{\pi}(s) = g(s)$$

Then, similarly to the infinite-horizon discounted case discussed in the main text, we have $V_T^* = g$ and for $t \in \{0, \ldots, T-1\}$,

$$Q_t^*(s, a) = f(t, s, a) + \gamma \mathbb{E}_{S' \sim \mathcal{T}(\cdot|s,a)} \Big[ V_{t+1}^*(S') \Big],$$

$$V_t^*(s) = V_t^{\pi^*}(s) = \lambda \log \sum_{a' \in \mathcal{A}} \exp \Big\{ Q_t^*(s, a')/\lambda \Big\},$$

$$\pi_t^*(a|s) = \exp \Big\{ Q_t^*(s, a)/\lambda \Big\} \Big/ \sum_{a' \in \mathcal{A}} \exp \Big\{ Q_t^*(s, a')/\lambda \Big\} = \exp \Big\{ \Big( Q_t^*(s, a) - V_t^*(s) \Big)/\lambda \Big\}.$$

Rearranging this system of equations, for any chosen function $v : \{0, ..., T\} \times \mathcal{S} \to \mathbb{R}$ with $v(T, \cdot) = g(\cdot)$, we see that $\pi_t^*(a|s)$ is the optimal strategy for the reward function

$$f(t, s, a) = \lambda \log \pi_t^*(a|s) - \gamma \sum_{s' \in \mathcal{S}} \mathcal{T}(s'|s, a)v(t+1, s') + v(t, s),$$

in which case the corresponding value function is $V^* = v$. In other words, the identifiability issue discussed earlier remains. We note that identifying $\pi$ in this setting is more delicate than in the infinite-horizon case, as it is necessary to observe many finite-horizon state-action trajectories, rather than a single infinite-horizon trajectory.

## 4.1 Time-homogeneous finite-horizon identifiability.

Following the release of a preprint version of this paper, Kim et al. [2021] was published and presented a closely related analysis, for entropy-regularized deterministic MDPs with zero terminal value. We here give an extension of their result which covers the stochastic case and includes an arbitrary (known) terminal reward.

The key structural assumptions made by Kim et al. [2021] are that the reward is time-homogeneous (that is, $f$ does not depend on $t$), and that there is a finite horizon. As discussed in the previous section, there is no guarantee that an arbitrary observed policy will be consistent with these assumptions (that is, whether there exists any $f$ generating the observed policy). However, given a policy consistent with these assumptions, and mild assumptions on the structure of the MDP, we shall see that unique identification of $f$ is possible up to a constant.

**Definition 2.** *We say a MDP has full access at horizon $T$ (from a state $s$) if, for some distribution over actions, for all states $s'$ we have $\mathbb{P}(S_{T-1} = s'|S_0 = s) > 0$.*

It is easy to verify that this definition does not depend on the choice of distribution over actions (provided it has full support).

This is slightly weaker than assuming that the Markov chain underlying the MDP (with random actions) is irreducible, as there may exist transient states from which we have full access. It is a classical result (commonly stated as a corollary to the Perron–Frobenius theorem) that an irreducible aperiodic Markov chain has full access (from every state). Kim et al. [2021] give an alternative graph-theoretic view, based on the closely related notion of $T$-coverings.

**Theorem 3.** *Consider an MDP with unknown time-homogeneous reward function $f$. In order for $f$ to be identified (up to a global constant) from observation of optimal policies and the resulting transitions up to some horizon $T > 0$, with initialization from some state $s$, it is necessary that the MDP has full access at some horizon $T' \geq T$ (from state $s$).*

The following definition is most easily expressed by associating our finite state space $\mathcal{S}$ with the set of basis vectors $\{e_k\}_{k=1}^N \subset \mathbb{R}^N$, and writing the transitions $\mathcal{T}(s'|s, a)$, for $a \in \mathcal{A}$ in terms of the transition matrix $\mathbb{T}(a)$ with

$$\mathbb{T}(a)_{ij} = \mathcal{T}(e_j|e_i, a).$$

While this definition is quite abstract, we will see that it precisely describes when many IRL problem with fixed terminal reward can be solved.

**Definition 3.** *We say an $N$-state MDP has full action-rank on horizon $T$, starting at a state $s \equiv e_i$, if the matrix with rows given by*

$$\left\{ e_i^\top \left( \sum_{t=0}^{T-1} \gamma^t \prod_{t'=0}^{t-1} \mathbb{T}(a_{t'}) \right); \qquad a_0, ..., a_t \in \mathcal{A} \right\}$$

*is of rank $N$ (with the convention that products are taken sequentially on the right, that is $\prod_{t=0}^2 A_t = A_0 A_1 A_2$, and the empty product is the identity).*

**Remark 6.** *Observe that $e_i^\top \prod_{t'=0}^{t-1} \mathbb{T}(a_{t'})$ is the expected state of $S_t$ given $S_0 = e_i$, when following the actions $\{a_0, ..., a_t\}$. Hence, the quantity $e_i^\top \left( \sum_{t=0}^{T-1} \gamma^t \prod_{t'=0}^{t-1} \mathbb{T}(a_{t'}) \right)$ is a time-weighted expected occupation density for the process, that is, a measurement of how long we spend in each state. We have full action-rank if our actions are sufficiently varied that there are $N$ linearly independent such density vectors (cf. [Kim et al., 2021, Corollary 1], where it is the state–action occupation density which is considered).*

**Theorem 4.** *Suppose our MDP has full action rank and full access, at horizon $T$, from an initial state $s_0$. Then the time-homogeneous IRL problem is identifiable, that is, knowledge of the (time-dependent) entropy-regularized optimal strategy $\pi_t^*(a|s)$, and the terminal reward $g$, is sufficient to uniquely determine a time-homogeneous running reward $f$, if it exists, up to a constant.*

*Conversely, if our MDP has full access but not full action rank at horizon $T$, from the state $s_0$, the IRL problem remains ill posed.*

As a corollary, we demonstrate a generalized version of [Kim et al., 2021, Theorem 2].

**Corollary 1.** *Suppose $\gamma \neq 0$ and our MDP is deterministic, that is $\mathcal{T}(s'|s, a) \in \{0, 1\}$, and one of the following holds:*

*(i) the underlying Markov chain is irreducible and aperiodic (i.e. with randomly chosen actions, the underlying Markov chain is irreducible and aperiodic)*

*(ii) the initial state $s_0 = e_i$ admits a self-loop (i.e. it is possible to transition from this state to itself), and all states can be accessed from the initial state in at most $d$ transitions*

*(iii) there exist cycles[7] starting at the initial state $s_0 = e_i$ with lengths $|Q|, |Q'|$, such that $\gcd(|Q|, |Q'|) = 1$, and all states can be accessed from the initial state in at most $d$ transitions.*

*Then there exists a horizon $T$ such that the time-homogeneous IRL problem is identifiable (as in Theorem 4). In particular, in case (ii), it is sufficient to take any finite $T \geq d + 1$; in case (iii) it is sufficient to take any finite $T \geq d + RR'$.*

We can extend this result to a stochastic setting, assuming that our action space is sufficiently rich.

**Corollary 2.** *Suppose $\gamma \neq 0$, and our MDP is stochastic and satisfies one of the sets of assumptions ((i), (iii) or (ii)) of Corollary 1 and that from every state, we have at least as many actions (with linearly independent resulting transition probabilities) as we have possible future states, that is,*

$$\text{rank}\{\mathcal{T}(\cdot|s, a); a \in \mathcal{A}\} = \#\{s' : \mathcal{T}(s'|s, a) > 0 \text{ for some } a \in \mathcal{A}\}.$$

*Then for any initial state $s_0$, there exists a horizon $T$ such that the time-homogeneous IRL problem is identifiable (as in Theorem 4). The sufficient bounds on $T$ from Corollary 1 also apply.*

This result also addresses a concern raised in Magnac and Thesmar [2002, Section 4.1], as it demonstrates that, for sufficiently rich control problems, a time-homogeneity assumption on rewards and the existence of a finite horizon is sufficient to guarantee identifiability.

## 5 Action-independent rewards

Earlier works such as Amin and Singh [2016], Amin et al. [2017], Dvijotham and Todorov [2010] and Fu et al. [2018] consider the case of action-independent rewards, that is, where $f$ is not a function of $a$. In general, it is not immediately clear whether, for a given observed policy, the IRL problem will admit an action-independent solution. In this section, we obtain a necessary and sufficient condition under which an action-independent time-homogeneous reward function could be a solution to a given entropy-regularized, infinite-time-horizon[8] IRL problem with discounting. We shall also obtain a rigorous condition under which a unique reward function can be identified.

Consider an entropy-regularized MDP environment $(\mathcal{S}, \mathcal{A}, \mathcal{T}, \gamma, \lambda)$, as given in Section 2. Without loss of generality, assume that $|\mathcal{S}|, |\mathcal{A}| \geq 2$ and $\mathcal{S} = \{s_1, \ldots, s_{|\mathcal{S}|}\}$. Let $\bar{\pi} : \mathcal{S} \to \mathcal{P}(\mathcal{A})$ be the observed optimal policy such that $\bar{\pi}(a|s) > 0$ for any $(s, a) \in \mathcal{S} \times \mathcal{A}$.

For $a \in \mathcal{A}$ we write $\bar{\pi}(a) \in \mathbb{R}^{|\mathcal{S}|}$ for the probability vector $\left(\bar{\pi}(a|s_1) \quad \ldots \quad \bar{\pi}(a|s_{|\mathcal{S}|})\right)^T$, and $\mathbb{T}(a) \in \mathbb{R}^{|\mathcal{S}| \times |\mathcal{S}|}$ for the transition matrix with $[\mathbb{T}(a)]_{ij} = \mathcal{T}(s_j|s_i, a)$. Fix a particular action $a_0 \in \mathcal{A}$, and write

$$\Delta \log \bar{\pi}(a) = \log \bar{\pi}(a) - \log \bar{\pi}(a_0) \quad \text{and} \quad \Delta \mathbb{T}(a) = \mathbb{T}(a) - \mathbb{T}(a_0),$$

where $\log \bar{\pi}(a)$ denotes the element-wise application of logarithm over the vector $\bar{\pi}(a)$, for any $a \in \mathcal{A}$.

---

[7]A cycle is a sequence of possible transitions which start and end in the same state. The length of a cycle is defined to be the number of transitions, e.g. a cycle $\{s_0 \to s_1 \to s_2 \to s_0\}$ has length 3. An irreducible Markov chain is aperiodic if there is no common factor (greater than one) of the lengths of all cycles.

[8]The analogous results for finite-horizon problems with time-inhomogeneous rewards (and general discount factor) can be obtained through the same method.

**Theorem 5.** *The above IRL problem admits a solution with action-independent reward $f : \mathcal{S} \to \mathbb{R}$ if and only if the system of equations*

$$\lambda \Delta \log \bar{\pi}(a) = \gamma \Delta \mathbb{T}(a) v, \quad \forall a \in \mathcal{A}, \tag{8}$$

*admits a solution $v \in \mathbb{R}^{|\mathcal{S}|}$. (Note that this is a system of $|\mathcal{A}| \times |\mathcal{S}|$ equations in $|\mathcal{S}|$ unknowns, so this is a non-trivial assumption.)*

**Corollary 3.** *Suppose $\gamma \in [0, 1)$. Assuming a solution to (8) exists, the IRL problem is identifiable (i.e. the true action-independent reward function can be inferred up to a constant shift) if and only if, writing $\mathcal{K}(a)$ for the kernel of $\Delta \mathbb{T}(a)$, we know that*

$$\{c\mathbf{1} : c \in \mathbb{R}\} = \bigcap_{a \in \mathcal{A} \setminus \{a_0\}} \mathcal{K}(a) = \bigcap_{a \in \mathcal{A}} \mathcal{K}(a),$$

*where $\mathbf{1}$ denotes the all-one vector in $\mathbb{R}^{|\mathcal{S}|}$. (Note that $\{c\mathbf{1} : c \in \mathbb{R}\} \subset \mathcal{K}(a)$ for any $a \in \mathcal{A} \setminus \{a_0\}$ and $\Delta \mathbb{T}(a_0) = 0$ implies $\mathcal{K}(a_0) = \mathbb{R}^{|\mathcal{S}|}$.)*

Under the entropy regularized framework, the long-run total reward depends on actions through the entropy penalty term. Therefore, it cannot be reduced to the scenario in Amin and Singh [2016], where any linear perturbation of the reward function will not affect optimal behavior under any given environment.

**Remark 7.** *Theorem 5 and Corollary 3 suggest various extensions, in the case when (8) does not admit a solution, but the assumed property on the kernels in Corollary 3 holds. For example, one could consider the least-squares solution to the system (8) (which is defined up to a constant). This gives a choice of value function which, in some sense, minimizes the action-dependence of the resulting cost function (obtained through Theorem 1).*

Fu et al. [2018] give a result similar to Corollary 3. Unfortunately, the role of the choice of actions in their conditions is not precisely stated, and on some interpretations is insufficient for the result to hold – as we have seen, the condition of Corollary 3 is both necessary and sufficient for identifiability. We give a variation of their assumptions in what follows.

**Definition 4** (Reward-decomposability)**.** *We say states $s_1, s_1'$ are '1-step linked', if there exist actions $a, a' \in \mathcal{A}$ and a state $s_0 \in \mathcal{S}$ such that $\mathcal{T}(s_1|s_0, a) > 0$ and $\mathcal{T}(s_1'|s_0, a') > 0$. We extend this definition through transitivity, forming a set of 'linked' states $\mathcal{S}_1$. We say say the MDP is reward-decomposable if all its states are linked.*

Note that there is no loss of generality if a specific $a'$ is selected in this definition (instead of being allowed to vary), provided $a$ is allowed to vary.

**Remark 8.** *An equivalent definition would be that our MDP is reward-decomposable if $\mathcal{S}_1 = \mathcal{S}$ is the only choice of nonempty set $\mathcal{S}_1 \subset \mathcal{S}$ such that: there exists a set $\mathcal{S}_0 \subset \mathcal{S}$ with (i) every transition (with any action) to $\mathcal{S}_1$ is from $\mathcal{S}_0$, and (ii) every transition from $\mathcal{S}_0$ is to $\mathcal{S}_1$. (In other words, $X_t \in \mathcal{S}_0$ if and only if $X_{t+1} \in \mathcal{S}_1$.) We note that Fu et al. [2018] simply call this property 'decomposable', but this seems an unfortunate choice of terminology given this alternative characterization.*

The following final corollary gives a simple set of conditions under which identification is possible, clarifying (and extending to the stochastic case) the result of [Fu et al., 2018, Theorem C.1].

**Corollary 4.** *Suppose our MDP either has deterministic transitions $\mathcal{T}(s'|s, a) \in \{0, 1\}$ or we have at least as many actions (with linearly independent resulting transition probabilities) as we have possible future states, that is,*

$$\text{rank}\{\mathcal{T}(\cdot|s, a); a \in \mathcal{A}\} = \#\{s' : \mathcal{T}(s'|s, a) > 0 \text{ for some } a \in \mathcal{A}\}.$$

*Then the (action-independent) IRL problem is identifiable (i.e. the true action-independent reward function can be inferred up to a constant shift) if and only if the MDP is reward-decomposable.*

It is clear that reward-decomposability is not, by itself, sufficient to guarantee identifiability of rewards – simply consider the trivial MDP with action space containing only one element (so no information can be gained by watching optimal policies) but all transitions are possible (so the MDP is reward-decomposable).

The necessity of reward-decomposability, in general, can easily be seen as follows: Suppose there are sets $\mathcal{S}_0, \mathcal{S}_1 \subset \mathcal{S}$ such that every transition from a state in $\mathcal{S}_0$ (under every action) is to a state in $\mathcal{S}_1$, and every transition to a state in $\mathcal{S}_1$ is from $\mathcal{S}_0$. Then, if we add $c \in \mathbb{R}$ to the reward in $\mathcal{S}_0 \setminus \mathcal{S}_1$, subtract $c/\gamma$ from the reward in $\mathcal{S}_1 \setminus \mathcal{S}_0$, and add $(1 - 1/\gamma)c$ to the reward in state $\mathcal{S}_0 \cap \mathcal{S}_1$, we will have no impact on the overall value or optimal strategies. A reward-decomposability assumption ensures $\mathcal{S}_1 = \mathcal{S}$ (which implies $\mathcal{S}_0 = \mathcal{S}$ as every transition into $\mathcal{S}_1$ must be from a state in $\mathcal{S}_0$), so this is simply a constant shift; otherwise, we see our IRL problem is not identifiable.

## Acknowledgements

The authors acknowledge the support of the Alan Turing Institute under the Engineering and Physical Sciences Research Council grant EP/N510129/1. Samuel Cohen also acknowledges the support of the Oxford-Man Institute for Quantitative Finance. As a visiting scholar, Haoyang Cao also appreciates the support provided by the Mathematical Institute at University of Oxford.

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
