# Identifiability in Inverse Reinforcement Learning:
## Supplementary Material

## A    Appendix: Proofs of Results

*Proof of Theorem 1.* Fix $f$ as in the statement of the theorem. Then (4) characterizes the corresponding value function

$$V_\lambda^*(s) = \lambda \log \sum_{a \in \mathcal{A}} \exp\left(\frac{1}{\lambda}\Big(f(s,a) + \gamma \sum_{s' \in \mathcal{S}} \mathcal{T}(s'|s,a)V_\lambda^*(s')\Big)\right)$$

$$= v(s) + \lambda \log \sum_{a \in \mathcal{A}} \bar{\pi}(a|s) \exp\left(\frac{\gamma}{\lambda}\Big(\sum_{s' \in \mathcal{S}} \mathcal{T}(s'|s,a)(V_\lambda^*(s') - v(s'))\Big)\right),$$

which rearranges to give

$$\exp(g(s)) = \sum_{a \in \mathcal{A}} \bar{\pi}(a|s) \exp\left(\gamma \sum_{s' \in \mathcal{S}} \mathcal{T}(s'|s,a)g(s')\right) \qquad (9)$$

with $g(s) = (V_\lambda^*(s) - v(s))/\lambda$. Applying Jensen's inequality, we can see that, for $\underline{s} \in \arg\min_{s \in \mathcal{S}} g(s)$,

$$\exp\left(\min_s g(s)\right) = \exp\left(g(\underline{s})\right) \geq \exp\left(\gamma \sum_{a \in \mathcal{A}, s' \in \mathcal{S}} \bar{\pi}(a|\underline{s})\mathcal{T}(s'|\underline{s},a)g(s')\right).$$

However, the sum on the right is a weighted average of the values of $g$, so

$$\sum_{a \in \mathcal{A}, s' \in \mathcal{S}} \bar{\pi}(a|\underline{s})\mathcal{T}(s'|\underline{s},a)g(s') \geq \min_s g(s).$$

Combining these inequalities, along with the fact $\gamma < 1$, we conclude that $g(s) \geq 0$ for all $s \in \mathcal{S}$.

Again applying Jensen's inequality to (9), for $\bar{s} \in \arg\max_{s \in \mathcal{S}} g(s)$ we have

$$\max_s \left\{ \exp\left(g(s)\right) \right\} = \exp\left(g(\bar{s})\right) \leq \sum_{a \in \mathcal{A}, s' \in \mathcal{S}} \bar{\pi}(a|\bar{s})\mathcal{T}(s'|\bar{s},a) \exp\left(\gamma g(s')\right).$$

As the sum on the right is a weighted average, we know

$$\sum_{a \in \mathcal{A}, s' \in \mathcal{S}} \bar{\pi}(a|\bar{s})\mathcal{T}(s'|\bar{s},a) \exp\left(\gamma g(s')\right) \leq \max_s \left\{ \exp\left(\gamma g(s)\right) \right\}.$$

Hence, as $\gamma < 1$, we conclude that $g(s) \leq 0$ for all $s \in \mathcal{S}$.

Combining these results, we conclude that $g \equiv 0$, that is, $V_\lambda^* = v$. Finally, we substitute the definition of $f$ and the value function $v$ into (6) to see that the entropy-regularized optimal policy is $\pi_\lambda^* = \bar{\pi}$.    $\square$

*Proof of Theorem 2.* From Theorem 1, if we can determine the value function for one of our agents, then the reward is uniquely identified. Given we know both agents' policies $(\pi, \tilde{\pi})$ and our agents are optimizing their respective MDPs, for every $a \in \mathcal{A}, s \in \mathcal{S}$, we know the value of

$$\lambda \log \frac{\pi(a|s)}{\tilde{\pi}(a|s)} = \gamma \sum_{s' \in \mathcal{S}} \mathcal{T}(s'|s,a)v(s') - \tilde{\gamma} \sum_{s' \in \mathcal{S}} \tilde{\mathcal{T}}(s'|s,a)v(s') - (v(s) - \tilde{v}(s)) \qquad (10)$$

where $v, \tilde{v}$ are the agents' respective value functions. This is an inhomogeneous system of linear equations in $\{v(s), \tilde{v}(s)\}_{s \in \mathcal{S}}$. Therefore, by standard linear algebra (in particular, the Fredholm alternative), it is uniquely determined up to the addition of solutions to the homogeneous equation

$$0 = \gamma \sum_{s' \in \mathcal{S}} \mathcal{T}(s'|s,a)v(s') - \tilde{\gamma} \sum_{s' \in \mathcal{S}} \tilde{\mathcal{T}}(s'|s,a)v(s') - (v(s) - \tilde{v}(s)) \text{ for all } s \in \mathcal{S}, a \in \mathcal{A}.$$

However, as we have assumed our pair of MDPs is value-distinguishing, the only solutions to this equation have at least one of $v$ and $\tilde{v}$ constant (we assume $v$ without loss of generality). Therefore, the space of solutions to (10) is either empty (in which case no consistent reward exists), or determines $v$ up to the addition of a constant. Given $v$ is determined up to a constant we can use Theorem 1 to determine $f$, again up to the addition of a constant. □

The following result[9] from elementary number theory will prove useful in what follows.

**Lemma 1.** *Let $\mathcal{R} \subset \mathbb{N}$ be a set of natural numbers, with the property that $\mathcal{R}$ is closed under addition (if $a, b \in \mathcal{R}$ then $a + b \in \mathcal{R}$). Suppose $\mathcal{R}$ has greatest common divisor $1$ (i.e. $\gcd(\mathcal{R}) = 1$). Then there exist elements $a, b \in \mathcal{R}$ which are coprime (i.e. $\gcd(a, b) = 1$). Furthermore, for any coprime $a, b \in \mathcal{R}$, for all $c \geq ab$, we know $c \in \mathcal{R}$, in particular, there exist at least two distinct pairs of nonnegative integers $\lambda, \mu$ such that $\lambda a + \mu b = c$.*

*Proof.* We first show a coprime pair $a, b \in \mathcal{R}$ exists. As $\gcd(\mathcal{R} \cap \{x : x \leq y\})$ is decreasing in $y$, and the integers are discrete, there exists a smallest value $y$ such that $\gcd(\mathcal{R} \cap \{x : x \leq y\}) = 1$. Applying Bézout's lemma, there exist integers $\{\lambda_k\}_{k \leq y}$ such that

$$\sum_{k \in \mathcal{R}, k \leq y} \lambda_k k = 1.$$

Rearranging this sum by taking all negative terms to the right hand side, we obtain the desired positive integers $a = \sum_{\{k \in \mathcal{R}, k \leq y, \lambda_k > 0\}} \lambda_k k$ and $b = \sum_{\{k \in \mathcal{R}, k \leq y, \lambda_k < 0\}} |\lambda_k| k$ which satisfy $a = b + 1$ (so $a$ and $b$ are coprime) and $a, b \in \mathcal{R}$ (as $\mathcal{R}$ is closed under addition).

We now take an arbitrary coprime pair $a, b \in \mathcal{R}$. Again by Bézout's lemma, there exist (possibly negative) integers $\tilde{\lambda}, \tilde{\mu}$ such that $\tilde{\lambda} a + \tilde{\mu} b = 1$, and hence $\tilde{\lambda} c a + \tilde{\mu} c b = c$. However, for any integer $k$ it follows that $(\tilde{\lambda} c + kb)a + (\tilde{\mu} c - ka)b = c$. Since this holds for all $k \in \mathbb{Z}$, we can choose $k$ such that $1 \leq \lambda = (\tilde{\lambda} c + kb) \leq b$. However, this implies that $(\tilde{\lambda} c + kb)a \leq ab \leq c$, and so $\mu = (\tilde{\mu} c - ka) \geq 0$. As $\mathcal{R}$ is closed under addition, we see that $c = \lambda a + \mu b \in \mathcal{R}$.

To see non uniqueness, we simply observe that if $c \geq ab$, in the construction above we have $\mu \geq a$, and hence $(\lambda + b, \mu - a)$ is an alternative pair of coefficients. □

*Proof of Theorem 3.* We suppose that $f$ can be identified, and first show that all states can be accessed from $s$, that is, for each $s'$ there exists $T > 0$ such that $\mathbb{P}^\pi(S_T = s'|S_0 = s) > 0$, but that $T$ can vary with $s'$. Suppose there were states which cannot be reached by a path starting in $s$. It is clear that it would be impossible to identify the cost associated with any state which cannot be accessed, as we obtain no information about actions in these states. Therefore, all states can be accessed from $s$.

It remains to show that, if $f$ can be identified, we can reach all states using paths of a common length. We initially focus on the paths from $s$ to $s$. If we can return in precisely $T$ steps, then (by the Markov property) we can also return in $kT$ steps, for any $k \in \mathbb{N}$. Therefore either the set $\{T : \mathbb{P}^\pi(S_T = s|S_0 = s) > 0\}$ is unbounded, or the state $s$ will never be revisited (in the language of Markov chains, it is ephemeral). If the starting state is ephemeral, it is clear that we can add a constant to its rewards independently of all other states' rewards, as this will not affect decision making – we leave this state immediately and never return. This would imply that the reward cannot be determined up to a global constant, and hence we conclude that the starting state is not ephemeral.

Next, still focusing on paths from $s$ to $s$, we show that $T$ can take *any* value above some bound. Let $\bar{t}$ be the greatest common divisor of $\mathcal{R} = \{t : \mathbb{P}^\pi(S_t = s|S_0 = s) > 0\}$; we will show that $\bar{t} = 1$. For contradiction, suppose $\bar{t} > 1$. Then our system is periodic, and by classical results on irreducible matrices (e.g. [Seneta, 2006, Theorem 1.3]) we know that there is a partition of $\mathcal{S}$ into $\bar{t}$ sets, such that we will certainly make transitions within the sets of states $\mathcal{S}_0 \to \mathcal{S}_1 \to ... \to \mathcal{S}_{\bar{t}-1} \to \mathcal{S}_0 \ni s$. By adding $c \in \mathbb{R}$ to the rewards of states in $\mathcal{S}_0$, and subtracting $c/\gamma$ from rewards of states in $\mathcal{S}_1$, we do not affect behavior. Therefore, the reward cannot be identified up to a global constant unless $\bar{t} = 1$.

However, given $\bar{t} = 1$ and the set $\mathcal{R}$ is closed under addition (by concatenating cycles), Lemma 1 implies that $\mathcal{R}$ must contain all sufficiently large values, that is, it is possible to return to the initial state in any sufficiently large number of steps.

---

[9]Thanks to Victor Flynn for discussion on the formulation and proof of this result.

Finally, we have seen that it is possible to transition from $s$ to $s'$ in a finite number of steps, and that it is possible to transition from $s$ to $s$ in any sufficiently large number of steps. From the Markov property we conclude that for every value of $T'$ sufficiently large, for all choices of $s'$ we have $\mathbb{P}^\pi(S_{T'} = s'|S_0 = s) > 0$. $\qquad\square$

*Proof of Theorem 4.* We first prove the sufficiency statement. The optimal policy satisfies

$$\lambda \log \pi_t^*(a|s) = Q_t^*(s, a) - V_t^*(s) = f(s, a) + \gamma\Big(\sum_{s'} \mathcal{T}(s'|s, a)V_{t+1}^*(s')\Big) - V_t^*(s).$$

We write (for notational simplicity), $\upsilon(s) = V_{T-1}^*(s)$, and hence, given $V_T^* \equiv g$ by assumption,

$$f(s, a) = \upsilon(s) + \lambda \log \pi_{T-1}^*(a|s) - \gamma\Big(\sum_{s'} \mathcal{T}(s'|s, a)g(s')\Big). \tag{11}$$

This shows that $f$ is completely determined (if it exists) by the function $\upsilon$.

We also observe that for every $t$ we have the recurrence relation

$$V_t^*(s) = -\lambda \log \pi_t^*(a|s) + f(s, a) + \gamma\Big(\sum_{s'} \mathcal{T}(s'|s, a)V_{t+1}^*(s')\Big)$$

$$= \lambda \log \frac{\pi_{T-1}^*(a|s)}{\pi_t^*(a|s)} + \upsilon(s) + \gamma\Big(\sum_{s'} \mathcal{T}(s'|s, a)\big(V_{t+1}^*(s') - g(s')\big)\Big).$$

This holds for any choice of action $a$ (unlike the usual dynamic programming relation, which only involves the optimal policy). Writing $\mathbf{V}_t$ for the vector with components $\{V_t^*(s)\}_{s\in\mathcal{S}}$ we have the recurrence relation

$$\mathbf{V}_t = \Upsilon_t(a) + \upsilon + \gamma\mathbb{T}(a)\mathbf{V}_{t+1}; \qquad \mathbf{V}_{T-1} = \upsilon, \tag{12}$$

where $\Upsilon_t$ is a known vector valued function, with components

$$[\Upsilon_t(a)]_s = \lambda \log \frac{\pi_{T-1}^*(a|s)}{\pi_t^*(a|s)} - \gamma\sum_{s'} \mathcal{T}(s'|s, a)g(s').$$

Solving the recurrence relation, we have, for any sequence of actions $a_0, ..., a_{T-1}$ (with the convention that the empty matrix product is the identity)

$$\mathbf{V}_0 = \Big(\sum_{t=0}^{T-1}\Big[\gamma^t\Big(\prod_{t'=0}^{t-1}\mathbb{T}(a_{t'})\Big)\Upsilon_t(a_t)\Big]\Big) + \Big(\sum_{t=0}^{T-1}\gamma^t\prod_{t'=0}^{t-1}\mathbb{T}(a_{t'})\Big)\upsilon + \gamma^T\Big(\prod_{t'=0}^{T-1}\mathbb{T}(a_{t'})\Big)g.$$

From this linear system, we can extract the single row corresponding to the fixed initial state $s_0$. Assuming this is the row indicated by the $e_i$ basis vector, we have

$$V_0^*(s_0) = e_i^\top\Big(\sum_{t=0}^{T-1}\gamma^t\prod_{t'=0}^{t-1}\mathbb{T}(a_{t'})\Big)\upsilon + G(a_0, ..., a_{T-1}) \tag{13}$$

for a known function $G$, expressible in terms of $\gamma$, $g$ and $\{\pi_t^*\}_{t=0}^{T-1}$.

Now that the MDP has full action-rank, the system of equations,

$$-G(a_0, \ldots, a_{T-1}) = e_i^\top\Big[\sum_{t=0}^{T-1}\gamma^t\prod_{t'=0}^{t-1}\mathbb{T}(a_{t'})\Big]\upsilon, \quad \forall a_0, \ldots, a_{T-1},$$

admits at most one solution, denoted by $\bar\upsilon$. Substituting into (13), we have a unique solution to the equation $V_0^*(s_0) = 0$. However, we need to consider all possible values of $V_0^*(s_0)$.

For any choice of actions $\{a_t\}_{t=0}^{T-1}$,

$$e_i^\top\Big[\sum_{t=0}^{T-1}\gamma^t\prod_{t'=0}^{t-1}\mathbb{T}(a_{t'})\Big]\mathbf{1} = \begin{cases} \frac{1-\gamma^T}{1-\gamma}, & \gamma \in (0, 1), \\ T, & \gamma = 1. \end{cases}$$

Here $\mathbf{1}$ denotes the all-one vector in $\mathbb{R}^N$. Therefore, the set of all possible $(V_0^*(s_0), \upsilon)$ pairs is given by

$$
\begin{cases}
\left\{ (c, \bar{\upsilon} + \frac{c(1-\gamma)}{1-\gamma^T}) : \forall c \in \mathbb{R} \right\}, & \gamma \in (0,1), \\
\left\{ (c, \bar{\upsilon} + \frac{c}{T}) : \forall c \in \mathbb{R} \right\}, & \gamma = 1.
\end{cases}
$$

From (11), we conclude that $f$ can be identified up to a constant.

To show necessity, we observe from the above that, if the system is not full action-rank, then there exists a linear subspace of choices of $\upsilon$, which do not differ only by constants, such that we can construct the same value vectors $\mathbf{V}_t$ for all $t$, satisfying (12) and hence (11). It follows that we have a nontrivial manifold of rewards $f$ which generate the same optimal policies, that is, the rewards are not identifiable. $\qquad\square$

*Proof of Corollary 1.* We first observe that it is a classical result on Markov chains (see, for example, [Seneta, 2006, Theorem 1.5]) that the conditions of case (i) guarantee those of case (iii), for some choice of $R, R' > 0$. The conditions of case (ii) also guarantee those of case (iii), with both the cycles being the self-loop. It is therefore sufficient to consider case (iii).

To show that the MDP has full action rank, we observe that for every possible path of states, there exists a corresponding sequence of actions, and vice versa. We will therefore use these different perspectives interchangeably. We also observe that, as our MDP is deterministic, $e_i^\top \prod_{t'=0}^{t-1} \mathbb{T}(a_{t'})$ is a vector indicating the current state at time $t$, when started in state $e_i$. Therefore,

$$
\mathbb{O}_{T-1}(\{a_t\}_{t \geq 0}) := e_i^\top \left( \sum_{t=0}^{T-1} \gamma^t \prod_{t'=0}^{t-1} \mathbb{T}(a_{t'}) \right)
$$

is a row vector, containing a time-weighted occupation density – in particular, if $\gamma = 1$, it simply counts the number of times we have entered each state. (This is in contrast to Remark 6, where we have an *expected* occupation density; here we can simplify given the control problem is deterministic.) Our aim, therefore, is to construct a collection of paths which give a full-rank system of occupation densities.

Starting in state $s_0 \equiv e_i$, consider a shortest path (i.e. a path with the fewest number of transitions) to each state $s'$. Denote these paths $r_{s'} = \{s_0 \to ... \to s'\}$, and the corresponding sequence of actions $a^{s'}$. These paths have lengths $|r_s|$ and time-weighted occupation densities $\mathbb{O}_{|r_s|}(\{a_t^s\}_{t \geq 0})$ which are linearly independent (a longer path will contain states not in a shorter path, while paths of the same length will differ in their final state; by reordering the states we can then obtain a lower-triangular structure in the matrix of occupation densities $[\mathbb{O}_{|r_s|}(\{a_t^s\}_{t \geq 0})]_{\{a_t\} \subset \mathcal{A}}$). This gives us $N = |\mathcal{S}|$ paths, of varying lengths, with linearly independent occupation densities.

We now consider prefixing our paths with cycles, in order to make them the same length. Fix an arbitrary integer value $T' \geq \max_s |r_s| + |Q||Q'| - 1$. By Lemma 1, for all states $s$, there exist nonnegative integers $\lambda_s, \mu_s$ such that $T' = \lambda_s |Q| + \mu_s |Q'| + |r_s|$. Therefore, taking the concatenated path consisting of $\lambda_s$ repeats of cycle $Q$, then $\mu_s$ repeats of cycle $Q'$, then our shortest path $r_s$, gives us a path from $s_0$ to $s$ of length $T'$. Denote each of these paths $P_s$.

Concatenation of paths has an elegant effect on the occupation densities: If $Q$ is a cycle and $r$ a path (starting from the terminal state of $Q$), their concatenation $Q * r$ and corresponding actions $a^Q, a^r, a^{Q*r}$, then the occupation densities combine linearly:

$$
\mathbb{O}_{|Q*r|}(\{a_t^{Q*r}\}_{t \geq 0}) = \mathbb{O}_{|Q|-1}(\{a_t^Q\}_{t \geq 0}) + \gamma^{|Q|} \mathbb{O}_{|r|}(\{a_t^r\}_{t \geq 0}), \tag{14}
$$

(observe that the occupation density excludes the (repeated) final state of the cycle).

We now observe that for the initial state, the shortest path is of length zero (i.e. has no transitions). From Lemma 1, as $T' \geq |Q||Q'|$, we know that there are multiple choices of $\lambda, \mu$ satisfying the stated construction, and therefore there are at least *two* possible paths $P_{s_0}$ and $\tilde{P}_{s_0}$ with the desired length, from the initial state to itself, using distinct numbers of cycles[10]: $(\lambda_{s_0}, \mu_{s_0})$ and $(\tilde{\lambda}_{s_0}, \tilde{\mu}_{s_0})$.

---

[10]If the cycles are both a self-loop, then this becomes degenerate, but in the following step the final column and row of the matrix $M$ can be omitted, and the remainder of the argument follows in essentially the same way.

This construction yields a collection of paths with full rank occupation densities. To verify this explicitly, we extract the rows corresponding to the paths $\{P_s\}_{s\in\mathcal{S}}$ and $\tilde{P}_{s_0}$, and use (14) to see that

$$
\begin{bmatrix}
\mathbb{O}_{T-1}(\{a_t^{P_{s_0}}\}_{t\geq 0}) \\
\mathbb{O}_{T-1}(\{a_t^{P_{s_1}}\}_{t\geq 0}) \\
\cdots \\
\mathbb{O}_{T-1}(\{a_t^{P_{s_N}}\}_{t\geq 0}) \\
\mathbb{O}_{T-1}(\{a_t^{\tilde{P}_{s_0}}\}_{t\geq 0})
\end{bmatrix}
= M
\begin{bmatrix}
\mathbb{O}_{|r_{s_0}|}(\{a_t^{r_{s_0}}\}_{t\geq 0}) \\
\mathbb{O}_{|r_{s_1}|}(\{a_t^{r_{s_1}}\}_{t\geq 0}) \\
\cdots \\
\mathbb{O}_{|r_{s_N}|}(\{a_t^{r_{s_N}}\}_{t\geq 0}) \\
\mathbb{O}_{|Q|-1}(\{a_t^{Q}\}_{t\geq 0}) \\
\mathbb{O}_{|Q'|-1}(\{a_t^{Q'}\}_{t\geq 0})
\end{bmatrix}
\tag{15}
$$

where

$$
\Gamma(\lambda, Q) :=
\begin{cases}
(1 - \gamma^{\lambda|Q|})/(1 - \gamma^{|Q|}), & \gamma \neq 1, \\
\lambda, & \gamma = 1,
\end{cases}
$$

$$
M =
\begin{bmatrix}
\gamma^{T'} & 0 & \cdots & 0 & \Gamma(\lambda_{s_0}, Q) & \gamma^{\lambda_{s_0}|Q|}\Gamma(\mu_{s_0}, Q') \\
0 & \gamma^{T'-|r_{s_1}|} & \cdots & 0 & \Gamma(\lambda_{s_1}, Q) & \gamma^{\lambda_{s_1}|Q|}\Gamma(\mu_{s_1} Q') \\
 & & \ddots & & & \\
0 & 0 & \cdots & \gamma^{T'-|r_{s_N}|} & \Gamma(\lambda_{s_N}, Q) & \gamma^{\lambda_{s_N}|Q|}\Gamma(\mu_{s_N} Q') \\
\gamma^{T'} & 0 & \cdots & 0 & \Gamma(\tilde{\lambda}_{s_0}, Q) & \gamma^{\tilde{\lambda}_{s_0}|Q|}\Gamma(\tilde{\mu}_{s_0}, Q')
\end{bmatrix}.
$$

After subtracting the first from the last row of $M$, as $\lambda_{s_0} \neq \tilde{\lambda}_{s_0}$, we see that $M$ has a simple structure, in particular it is a full-rank matrix with $N + 1$ rows and $N + 2$ columns. As the final matrix on the right hand side of (15) is of rank $N$, this implies that the left hand side of (15) is also of rank $N$ (by Sylvester's rank inequality). As the left hand side of (15) is a selection of rows from the matrix considered in Definition 3, we conclude that our MDP must be of full action rank.

Our collection of paths also shows that our system has full access at horizon $T = T' + 1$, and therefore the identification result follows from Theorem 4. By varying $T'$, we see this result holds for any choice of $T \geq |Q||Q'| + \max_s |r_s|$, as desired. $\qquad\square$

**Example 1.** *Consider the problem with three states $\mathcal{S} = \{A, B, C\}$, with possible transitions $A \rightarrow \{B, C\}$, $B \rightarrow A$ and $C \rightarrow B$. Starting in state $A$, the shortest paths are then given by $\{A\}, \{A \rightarrow B\}, \{A \rightarrow C\}$, and we have cycles $\{A \rightarrow B \rightarrow A\}$ and $\{A \rightarrow C \rightarrow B \rightarrow A\}$. Writing out the occupation densities of each of these paths (ignoring the terminal state of the two cycles), with $\gamma = 1$, we get the system*

$$
\begin{aligned}
\textit{Shortest paths} &\left\{ \\
\textit{Cycles (excluding final state)} &\left\{
\end{aligned}
\right.
\begin{bmatrix}
A \\
A \rightarrow B \\
A \rightarrow C \\
\hline
A \rightarrow B \\
A \rightarrow C \rightarrow B
\end{bmatrix}
\Rightarrow
\begin{array}{c}
(A \quad B \quad C) \\
\begin{bmatrix}
1 & 0 & 0 \\
1 & 1 & 0 \\
1 & 0 & 1 \\
\hline
1 & 1 & 0 \\
1 & 1 & 1
\end{bmatrix}
\end{array}.
$$

*This corresponds to the final term on the right hand side of (15). Clearly, the section above the horizontal line (corresponding to the shortest paths) is lower-triangular, and hence of full rank. We prefix our paths by appropriate numbers of cycles, in order to make them the same length. This implies that, with a horizon $T = 7 = 2 \times 3 + 1$, we consider the paths*

$$
\begin{bmatrix}
A \rightarrow C \rightarrow B \rightarrow A \rightarrow C \rightarrow B \rightarrow A \\
A \rightarrow B \rightarrow A \rightarrow C \rightarrow B \rightarrow A \rightarrow B \\
A \rightarrow B \rightarrow A \rightarrow C \rightarrow B \rightarrow A \rightarrow C \\
A \rightarrow B \rightarrow A \rightarrow B \rightarrow A \rightarrow B \rightarrow A \\
\cdots
\end{bmatrix}
\Rightarrow
\begin{bmatrix}
3 & 2 & 2 \\
3 & 3 & 1 \\
3 & 2 & 2 \\
4 & 3 & 0 \\
\cdots
\end{bmatrix}
= \left[ e_A^\top \left( \sum_{t=0}^{T-1} \gamma^t \prod_{t'=0}^{t-1} \mathbb{T}(a_{t'}) \right) \right]_{\{a_t\} \subset \mathcal{A}}
$$

*The matrix of occupation densities shown here is the left hand side of (15) and is easily seen to be full rank; the matrix $M$ from (15) is given by*

$$
M =
\begin{bmatrix}
1 & 0 & 0 & 0 & 2 \\
0 & 1 & 0 & 1 & 1 \\
0 & 0 & 1 & 1 & 1 \\
1 & 0 & 0 & 3 & 0
\end{bmatrix}.
$$

*Proof of Corollary 2.* We represent states by their corresponding basis vectors. For any initial state $e_i$, consider the space spanned by the possible future states. Given we have as many actions as possible future states, and the rank-nullity theorem, we know that this space must be the same as the space spanned by the vectors $\{\mathcal{T}(\cdot|s, a); a \in \mathcal{A}\}$. In particular, for any $e_j$ such that it is possible to transition from $e_i$ to $e_j$ in a single step, there exists a set of weights $c_a$ over actions (which do not need to sum to one or be nonnegative) such that $e_j = \sum_{a \in \mathcal{A}} c_a e_i^\top \mathbb{T}(a)$. In other words, there is no difference between the linear span generated by these stochastic transitions and deterministic transitions. As actions at every time can be varied independently, and the requirement that a MDP has full action rank depends only on the space spanned by transition matrices, the problem reduces to the setting of Corollary 1. □

*Proof of Theorem 5.* **Necessity.** Suppose the IRL problem admits an action-independent solution $f : \mathcal{S} \to \mathbb{R}$. Then for any $(s, a) \in \mathcal{S} \times \mathcal{A}$,

$$f(s) = \lambda \log \bar{\pi}(a|s) - \gamma \sum_{s' \in \mathcal{S}} \mathcal{T}(s'|s, a)v(s') + v(s),$$

where $v$ is the corresponding value function. Notice that for any $a \in \mathcal{A}$, for all $s \in \mathcal{S}$,

$$\begin{aligned} f(s) &= \lambda \log \bar{\pi}(a|s) - \gamma \sum_{s' \in \mathcal{S}} \mathcal{T}(s'|s, a)v(s') + v(s) \\ &= \lambda \log \bar{\pi}(a_0|s) - \gamma \sum_{s' \in \mathcal{S}} \mathcal{T}(s'|s, a_0)v(s') + v(s). \end{aligned}$$

Therefore, taking $v$ to be the vector with components $v(s)$, we have a solution to the system of equations (8).

**Sufficiency.** Let $v$ be a solution to the system of equations (8). By abuse of notation, we may write $v(s)$ for the components of $v$. Then for any $(s, a) \in \mathcal{S} \times \mathcal{A}$,

$$\lambda \log \bar{\pi}(a|s) - \gamma \sum_{s' \in \mathcal{S}} \mathcal{T}(s'|s, a)v(s') = \lambda \log \bar{\pi}(a_0|s) - \gamma \sum_{s' \in \mathcal{S}} \mathcal{T}(s'|s, a_0)v(s').$$

Therefore, the quantity $\hat{f}(s) := \lambda \log \bar{\pi}(a_0|s) - \gamma \sum_{s' \in \mathcal{S}} \mathcal{T}(s'|s, a_0)v(s') + v(s)$ is independent of $a$. From Theorem 1, we conclude that $\hat{f}$ is a solution to the IRL problem.

□

*Proof of Corollary 3.* Let $v_0$ be a solution to (8), which is assumed to exist. By the Fredholm alternative (as in Theorem 2) the solution set $\mathbb{Y}_\mathcal{S}$ for (8) is given by

$$\mathbb{Y}_\mathcal{S} = \left\{ v_0 + \kappa : \kappa \in \mathrm{span}\left( \bigcap_{a \in \mathcal{A}} \mathcal{K}(a) \right) \right\}.$$

From Theorem 5, the set of action-independent solutions for the IRL is given by

$$\mathbb{F}_\mathcal{S} = \left\{ f : f(s) = \lambda \log \bar{\pi}(a_0|s) - \gamma \sum_{s' \in \mathcal{S}} \mathcal{T}(s'|s, a_0)v(s') + v(s); \quad \text{for } v \in \mathbb{Y}_\mathcal{S}, s \in \mathcal{S} \right\}.$$

We then observe that the stated condition is sufficient – if constant vectors are the only valid choices for $\kappa$, then $v$ and hence $f \in \mathbb{F}_\mathcal{S}$ will only vary by constants.

To show necessity, denote by $f_0$ the solution corresponding to $v_0$. Suppose there exists a vector

$$\hat{v} \in \left( \bigcap_{a \in \mathcal{A}} \mathcal{K}(a) \right) \setminus \{c\mathbf{1} : c \in \mathbb{R}\}.$$

Define

$$\Delta(s) = \hat{v}(s) - \gamma \sum_{s' \in \mathcal{S}} \mathcal{T}(s'|s, a_0)\hat{v}(s'), \quad \forall s \in \mathcal{S}.$$

It follows that $f_0 + \Delta \in \mathbb{F}_\mathcal{S}$; if $\Delta$ is not a constant, we see that the reward is not uniquely identifiable.

To show $\Delta$ is not a constant, let

$$\overline{v} = \max_{s \in \mathcal{S}} \hat{v}(s), \quad \overline{s} \in \arg\max_{s \in \mathcal{S}} \hat{v}(s), \quad \underline{v} = \min_{s \in \mathcal{S}} \hat{v}(s), \quad \underline{s} = \arg\min_{s \in \mathcal{S}} \hat{v}(s), \quad \tilde{v} = \frac{\sum_{s \in \mathcal{S}} \hat{v}(s)}{|\mathcal{S}|}.$$

Then $\underline{v} = \hat{v}(\underline{s}) < \tilde{v} < \overline{v} = \hat{v}(\overline{s})$. We have

$$\Delta(\overline{s}) - (1 - \gamma)\tilde{v} = \overline{v} - \tilde{v} - \gamma \sum_{s \in \mathcal{S}} \mathcal{T}(s|\overline{s}, a_0)[\hat{v}(s) - \tilde{v}] \geq (1 - \gamma)(\overline{v} - \tilde{v}) > 0;$$

$$\Delta(\underline{s}) - (1 - \gamma)\tilde{v} = \underline{v} - \tilde{v} - \gamma \sum_{s \in \mathcal{S}} \mathcal{T}(s|\underline{s}, a_0)[\hat{v}(s) - \tilde{v}] \leq (1 - \gamma)(\overline{v} - \tilde{v}) < 0.$$

Therefore, $\Delta$ is not a constant. It follows that our condition is necessary in order to have an identifiable action-independent reward $\qquad\square$

*Proof of Corollary 4.* We will verify the condition of Corollary 3.

In the stochastic transition case, the proof of Corollary 2 shows that, under the stated assumption on the rank of the transitions, we can perform row operations on our transition matrix (corresponding to linear combinations of actions) to obtain a deterministic transition matrix. In particular, the space $\cap_{a \in \mathcal{A}} \mathcal{K}(a)$ is the same under the assumption on the rank of the transitions as under the assumption that transitions are deterministic. We can therefore focus our attention on the deterministic transition case.

If transitions are deterministic, the matrix $\mathbb{T}(a)$ has rows given by the basis vectors indicating the future states; so the matrix $\Delta\mathbb{T}(a)$ has rows which are the difference of two basis vectors corresponding to one-step linked states. Therefore, a vector $v \in \mathcal{K}(a)$ must have entries $v_i = v_j$ whenever $e_i$ and $e_j$ correspond to these one-step-linked states.

By considering all possible choices of $a$, we see that a vector $v \in \cap_{a \in \mathcal{A}} \mathcal{K}(a)$ must have entries $v_i = v_j$ whenever $e_i$ and $e_j$ correspond to any one-step linked states (and this is a sufficient condition to ensure $v \in \cap_{a \in \mathcal{A}} \mathcal{K}(a)$). However, if our MDP is reward-decomposable, there is no proper subset $\mathcal{S}_1$ of $\mathcal{S}$ which is closed under taking one-step linked states. Therefore, if our MDP is reward-decomposable, the only vectors in the kernel of $\Delta\mathbb{T}(a)$ for every $a$ are the constant vectors, as desired.

Conversely, if our MDP is not reward-decomposable, then there exists a set $\mathcal{S}_1 \neq \mathcal{S}$ satisfying the conditions above, and hence a nonconstant vector $v \in \cap_{a \in \mathcal{A}} \mathcal{K}(a)$. The result of Corollary 3 then shows the reward is not identifiable. $\qquad\square$

# B Appendix: A linear-quadratic-Gaussian problem

In this appendix, we will present the corresponding results for a class of one-dimensional linear-quadratic problems with Gaussian noise, ultimately inspired by Kalman [1964]. This simplified framework allows us to explicitly observe the degeneracy of inverse reinforcement learning, even if we add restrictions on the choice of value functions.

**Optimal LQG control** Suppose our agent seeks to control, using a real-valued process $A_t$ a discrete time process with dynamics

$$S_{t+1} = (\bar{\mu} + \mu_s S_t + \mu_a A_t) + (\bar{\sigma} + \sigma_s S_t + \sigma_a A_t) Z_{t+1}$$

for constants $\bar{\mu}, \mu_s, \mu_a, \bar{\sigma}, \sigma_s, \sigma_a$. The innovations process $Z$ is a Gaussian white noise with unit variance. Our agent uses a randomized strategy $\pi(a|s)$ to maximize the expectation of the entropy-regularized infinite-horizon discounted linear-quadratic reward:

$$\mathbb{E}\bigg[\sum_{t=1}^{\infty} \gamma^t \bigg(\int_{\mathbb{R}} f(S_t, a)\pi(a|s)da + \lambda\mathcal{H}(\pi(\cdot|S_t))\bigg)\bigg]$$

where $f(s, a) = \alpha_{20}s^2 + \alpha_{11}sa + \alpha_{02}a^2 + \alpha_{10}s + \alpha_{01}a + \alpha_{00}$ and $\mathcal{H}(\pi) = -\int_{\mathbb{R}} \pi(a)\log(\pi(a)da$ is the Shannon entropy of $\pi$. We assume the coefficients of $f$ are such that the problem is well posed (i.e. it is not possible to obtain an infinite expected reward).

Just as in the discrete state and action space setting, we can write down the state-action value function

$$Q_\lambda^\pi(s, a) = f(s, a) + \gamma \int_{\mathbb{R}} V_\lambda^\pi(s') \frac{1}{\sqrt{2\pi(\bar\sigma + \sigma_s s + \sigma_a a)^2}} \exp\Big( -\frac{(s' - \mu_s s - \mu_a a)^2}{2(\bar\sigma + \sigma_s s + \sigma_a a)^2} \Big) ds'. \quad (16)$$

Using this, the optimal policy and value function are given by

$$\pi_\lambda^*(a|s) = \exp\Big( (Q_\lambda^{\pi_\lambda^*}(s, a) - V_\lambda^{\pi_\lambda^*}(s))/\lambda \Big), \quad (17)$$

$$V_\lambda^*(s) = V_\lambda^{\pi_\lambda^*}(s) = \lambda \log \int_{\mathbb{R}} \exp\Big( Q_\lambda^{\pi_\lambda^*}(s, a)/\lambda \Big) da. \quad (18)$$

What is particularly convenient about this setting is that $Q$ is a quadratic in $(s, a)$, $V_\lambda$ is a quadratic in $s$, and $\pi_\lambda^*(\cdot|s)$ is a Gaussian density. In particular, the optimal policy is of the form

$$\begin{aligned}
\pi_\lambda^*(a|s) &= \frac{1}{\sqrt{2\pi\lambda k_3}} \exp\left\{ -\frac{(a - k_1 s - k_2)^2}{2\lambda k_3} \right\} \\
&= \exp\left\{ -\frac{1}{\lambda} \left[ \frac{a^2 - 2(k_1 s + k_2)a}{2k_3} + \frac{(k_1 s + k_2)^2}{2k_3} + \frac{\lambda}{2} \log(2\pi k_3 \lambda) \right] \right\}
\end{aligned} \quad (19)$$

for some constants $k_1, k_2 \in \mathbb{R}, k_3 > 0$, which can be determined[11] in terms of the known parameters $\mu_a, \mu_s, \sigma, \lambda$ and the parameters of the reward function $\{\alpha_{ij}\}_{i+j\le 2}$.

**Theorem 6.** *Consider an agent with a policy of the form* (19). *Suppose we also know that the value function $V$ is a quadratic (or, equivalently, that the reward function is a quadratic in $(s, a)$). The space of rewards consistent with this policy is given by:*

$$\begin{aligned}
\mathbb{F} = \Big\{ & f(s, a) = a_{20} s^2 + a_{11} sa + a_{02} a^2 + a_{10} s + a_{01} a + a_{00} \, \Big| \\
& (a_{20}, a_{11}, a_{02}) = \Big( \frac{-k_1^2}{2k_3}, \frac{k_1}{k_3}, \frac{-1}{2k_3} \Big) - \beta_2 \Big( \gamma(\mu_s^2 + \sigma_s^2) - 1, \, 2\gamma(\mu_s \mu_a + \sigma_s \sigma_a), \, \gamma(\mu_a^2 + \sigma_a^2) \Big), \\
& (a_{10}, a_{01}) = \Big( \frac{-k_1 k_2}{k_3}, \frac{k_2}{k_3} \Big) - \beta_2 \Big( 2\gamma(\bar\mu \mu_s + \bar\sigma \sigma_s), 2\gamma(\bar\mu \mu_a + \bar\sigma \sigma_a) \Big) - \beta_1 \Big( \gamma\mu_s + 1, \, \gamma\mu_a \Big), \\
& a_{00}, \beta_2, \beta_1 \in \mathbb{R} \Big\}.
\end{aligned} \quad (20)$$

*Proof.* We consider an arbitrary quadratic

$$v(s) = \beta_2 s^2 + \beta_1 s + \beta_0$$

as a candidate value function. If we have begun from the assumption that the reward function $f$ is quadratic, we know that the corresponding value function is quadratic, so this is not a restrictive assumption.

We then compute the state-action value function using (16), to give

$$Q_\lambda(s, a) = f(s, a) + \gamma\Big( \beta_2\big( (\bar\mu + \mu_s s + \mu_a a)^2 + (\bar\sigma + \sigma_s s + \sigma_a a)^2 \big) + \beta_1(\bar\mu + \mu_s s + \mu_a a) + \beta_0 \Big).$$

Combining with (19) and (17), we see that a reward function $f$ is consistent with the observed policy if

$$\begin{aligned}
\lambda \log \pi(a|s) &= -\left[ \frac{a^2 - 2(k_1 s + k_2)a}{2k_3} + \frac{(k_1 s + k_2)^2}{2k_3} + \frac{\lambda}{2} \log(2\pi k_3 \lambda) \right] \\
&= f(s, a) + \gamma\Big( \beta_2\big( (\bar\mu + \mu_s s + \mu_a a)^2 + (\bar\sigma + \sigma_s s + \sigma_a a)^2 \big) + \beta_1(\bar\mu + \mu_s s + \mu_a a) + \beta_0 \Big) \\
&\quad - \Big( \beta_2 s^2 + \beta_1 s + \beta_0 \Big).
\end{aligned}$$

---

[11]The explicit formulae for $k_1, k_2$ and $k_3$, and the coefficients of the value function, can be obtained by equating the coefficients of $f$ with the values obtained in (21). Under the assumption that the optimal control problem is well posed, this has a solution with $k_3 > 0$.

Rearranging, we conclude that $f$ is given by

$$
\begin{aligned}
f(a, s) = & \left[ -\frac{k_1^2}{2k_3} - \beta_2(\gamma(\mu_s^2 + \sigma_s^2) - 1) \right] s^2 + \left[ \frac{k_1}{k_3} - 2\beta_2\gamma(\mu_s\mu_a + \sigma_s\sigma_a) \right] as \\
& + \left[ -\frac{1}{2k_3} - \beta_2\gamma(\mu_a^2 + \sigma_a^2) \right] a^2 + \left[ -\frac{k_1 k_2}{k_3} - 2\beta_2\gamma(\bar{\mu}\mu_s + \bar{\sigma}\sigma_s) - \beta_1(\gamma\mu_s + 1) \right] s \\
& + \left[ \frac{k_2}{k_3} - 2\beta_2\gamma(\bar{\mu}\mu_a + \bar{\sigma}\sigma_a) - \beta_1(\gamma\mu_a) \right] a \\
& + \left[ -\frac{\lambda}{2}\log(2\pi k_3\lambda) - \beta_2\gamma(\bar{\mu}^2 + \bar{\sigma}^2) - \beta_1\bar{\mu} + \beta_0(1 - \gamma) \right].
\end{aligned}
\tag{21}
$$

As $(\beta_2, \beta_1, \beta_0)$ are arbitrary, we have the desired statement. $\qquad\square$

As in Theorem 1, we see that the inverse reinforcement learning problem only defines the rewards up to the choice of value function, which is arbitrary; the restriction to quadratic rewards or values simply reduces our problem to the smaller range of rewards determined by the three coefficients in the quadratic $V$.

The following theorem gives the linear-quadratic version of Theorem 2. As our agents' actions have a linear effect on the state variable, this leads to a particularly simple set of conditions for identifiability of the reward, given observation of two agents' policies.

**Theorem 7.** *Suppose we now have two agents, who are both following their respective optimal controls of the form* (19)*, for the same reward function, but disagree on some combination of the dynamics and discount rate. We write*

$$
x_1 = \left( \gamma\mu_s + 1, \ \gamma\mu_a \right) \qquad and \qquad x_2 = \left( \gamma(\mu_s^2 + \sigma_s^2) - 1, \ 2\gamma(\mu_s\mu_a + \sigma_s\sigma_a), \ \gamma(\mu_a^2 + \sigma_a^2) \right),
$$

*giving us two pairs of vectors $(x_1, x_2)$ (for the first agent) and $(\tilde{x}_1, \tilde{x}_2)$ (for the second agent). We assume we know these vectors for each agent. The quadratic reward function $f$ consistent with both agents' policies, if it exists, is uniquely identified up to the addition of a constant shift, if (and only if)*

$$
\frac{x_1}{\|x_1\|} \neq \frac{\tilde{x}_1}{\|\tilde{x}_1\|} \qquad and \qquad \frac{x_2}{\|x_2\|} \neq \frac{\tilde{x}_2}{\|\tilde{x}_2\|}.
$$

*Proof.* We see from (20) that a single agent's actions identify a space of valid rewards $\mathbb{F}$, which is parameterized by the constant shift $a_{00}$ and the two free variables $\beta_1, \beta_2$. From these free variables, (20) identifies the values of $\mathbf{a} = (a_{20}, a_{11}, a_{02}, a_{10}, a_{01})$. The reward function $f$ is uniquely defined, up to a constant shift, if we can identify the value of $\mathbf{a}$, which (by assumption) is the same for both agents.

Considering the role of $\beta_2$, (20) defines a line in $\mathbb{R}^3$ of possible values for $(a_{20}, a_{11}, a_{02})$. If the assumption $x_2/\|x_2\| \neq \tilde{x}_2/\|\tilde{x}_2\|$ holds, then the lines for our two agents will not be parallel, therefore will either never meet (in which case no consistent reward exists), or will meet at a point, uniquely identifying $(a_{20}, a_{11}, a_{02})$ and the corresponding values of $\beta_2$ for each agent. Conversely, if the assumption does not hold, then the lines will be parallel, so cannot meet in a unique point, in which case there are either zero or infinitely many reward functions consistent with both agents' policies.

Essentially the same argument then applies to the equation for $(a_{10}, a_{01})$. Given that $\beta_2$ has already been identified for each agent, varying $\beta_1$ for each agent defines a pair of lines in $\mathbb{R}^2$, which are not parallel if and only if the stated assumption on $x_1, \tilde{x}_1$ holds. Therefore, we can uniquely identify $(a_{10}, a_{01})$ if and only if the stated assumption holds. $\qquad\square$

Due to the simplicity of the characterization in Theorem 7, we can easily see that it is enough to observe two agents using different discount rates.

**Corollary 5.** *Suppose we observe two agents, each using optimal policies of the form* (19)*, for the same dynamics and rewards, but different discount rates. Then the underlying quadratic reward consistent with both agents' policies is identifiable up to a constant.*

*Proof.* Simply observe that the value of $\gamma$ introduces a non-scaling change in the vectors $x_1, x_2$ defined in Theorem 7. $\qquad\square$

We can also easily determine the identifiability of action-independent rewards.

**Corollary 6.** *For an agent with a policy of the form* (19)*, there exists an action-independent reward function corresponding to this policy if and only if*

$$k_1 = -\frac{\mu_s\mu_a + \sigma_s\sigma_a}{\mu_a^2 + \sigma_a^2}$$

*and this case, the action-independent reward is unique.*

*Proof.* From Theorem 6, in order to have an action independent reward we must have $a_{11} = a_{02} = a_{01} = 0$. From (20), we know

$$a_{11} = 0 \quad \Rightarrow \quad \beta_2 = \frac{k_1}{2k_3\gamma(\mu_s\mu_a + \sigma_s\sigma_a)}$$

$$a_{02} = 0 \quad \Rightarrow \quad \beta_2 = \frac{-1}{2k_3\gamma(\mu_a^2 + \sigma_a^2)}.$$

The statement $k_1 = -(\mu_s\mu_a + \sigma_s\sigma_a)/(\mu_a^2 + \sigma_a^2)$ is easily seen to be equivalent to stating that these equations are consistent.

The value of $\beta_1$ can then always be chosen in a unique way to guarantee $a_{01} = 0$, as required. $\qquad\square$

## C   Appendix: A discussion of guided cost learning and related maximum entropy inverse reinforcement learning models

The guided cost learning algorithm was proposed in Finn et al. [2016b] to solve an (undiscounted) inverse reinforcement learning problem over a finite time horizon with a finite state-action space $(\mathcal{S}, \mathcal{A})$. In Finn et al. [2016b], instead of directly modelling the optimal feedback policy, the optimal trajectory distribution is taken as the starting point for inference. Adopting the idea of the maximum casual entropy model in Ziebart [2010] (phrased in terms of rewards rather than costs) a common interpretation of the algorithm assumes we observe trajectories $\tau$ sampled from the distribution

$$p^f(\tau = (s_0^\tau, a_0^\tau, \ldots, s_{T-1}^\tau, a_{T-1}^\tau, s_T^\tau)) = \frac{1}{Z^f}\exp\left\{\sum_{t=0}^{T-1} f(s_t^\tau, a_t^\tau)\right\}, \tag{22}$$

where the partition factor

$$Z^f = \sum_\tau \exp\left\{\sum_{t=0}^{T-1} f(s_t^\tau, a_t^\tau)\right\} = \mathbb{E}_{\tau \sim q}\left[\exp\left\{\sum_{t=0}^{T-1} f(s_t^\tau, a_t^\tau)\right\}\middle/ q(\tau)\right]$$

is estimated through importance sampling with the 'ambient distribution' $q(\tau)$, which can be chosen arbitrarily[12].

As mentioned above, and discussed further by Ziebart et al. [2008] and Levine [2018], this is consistent with our entropy regularized MDP when transitions are deterministic, but differs for stochastic problems. An alternative maximum entropy model, which incorporates knowledge of $\mathcal{T}$, assumes trajectories are sampled from

$$\bar{p}^f(\tau) = \frac{\mu_0(s_0^\tau)}{Z^f}\prod_{t=0}^{T-1}\exp\left\{f(s_t^\tau, a_t^\tau)\right\}\mathcal{T}(s_{t+1}^\tau|s_t^\tau, a_t^\tau). \tag{23}$$

---

[12]An additional complexity in the guided cost learning algorithm is that the reward function and the ambient distribution are updated iteratively. Numerically, this can be seen as a variance reduction technique, rather than a conceptual change to the algorithm. First, the reward function $f$ is updated by alternately maximizing the log likelihood $\log p^f(\tau)$ over the demonstrator's trajectories $\{\tau_i^*\}_{i=1}^N$, which is equivalent to solving $\hat{f} = \arg\min_f D_{\text{KL}}(q^*\|p^f)$. Secondly, the ambient distribution $q$ is updated by minimizing the KL divergence $D_{KL}(q\|p^f)$ using the trajectories $\{\tau_j^q\}_{j=1}^M$ sampled from $q(\tau) = \mu_0(s_0^\tau)\prod_{t=0}^{T-1}\pi_t(a_t^\tau|s_t^\tau)\mathcal{T}(s_{t+1}^\tau|s_t^\tau, a_t^\tau)$. Using this method, the transition probabilities $\mathcal{T}$ can also be estimated, and $q$ can be seen as closely related to the law $\bar{p}^f$ in (23).

To see how this connects to the entropy regularized MDP, we observe that a entropy-regularized optimizing agent will generate trajectories with distribution

$$q^*(\tau) = \mu_0(s_0^\tau) \prod_{t=0}^{T-1} \pi_t^*(a_t^\tau | s_t^\tau) \mathcal{T}(s_{t+1}^\tau | s_t^\tau, a_t^\tau), \tag{24}$$

where $\pi^* = \{\pi_t^*\}_{t=0}^{T-1}$ solves the problem discussed in Section 4.

Given that we do not have an infinite-horizon time-homogenous system, the optimal policy $\pi^*$ is typically time-dependent and this is reflected in the density $q^*$, and hence in the trajectories we observe. Using $\bar{p}^f$ in (23) as the basis of the guided cost learning algorithm, the demonstrator's optimal trajectory distribution $q^*$ can be written in the desired form (i.e. for some choice of $f$ in (23), which may or may not correspond to the agent's rewards), provided the underlying $f_{\text{true}}$ and $g_{\text{true}}$ lead to a time-invariant optimal policy $\pi^*$. Otherwise, one should further adjust the guided cost learning model $\bar{p}^f$ to include time-dependent rewards $f$, that is,

$$\tilde{p}^f(\tau) = \frac{\mu_0(s_0^\tau)}{Z^f} \prod_{t=0}^{T-1} \exp\left\{f(t, s_t^\tau, a_t^\tau)\right\} \mathcal{T}(s_{t+1}^\tau | s_t^\tau, a_t^\tau). \tag{25}$$

With the addition of time-dependent rewards, it is interesting to consider what this variation of guided cost learning will output. Suppose we observe a large number of trajectories and estimate a reward $f_{\text{est}}$ to maximize the likelihood (25), or equivalently to minimize the KL divergence $D_{\text{KL}}(q^* \| \tilde{p}^f)$. Comparing $\tilde{p}^f$ in (25) with $q^*$ in (24), we see that the minimum KL divergence is $D_{\text{KL}}(q^* \| \tilde{p}^f) = 0$, which is achieved when, for each $t \in \{0 \ldots, T-1\}$,

$$f_{\text{est}}(t, s, a) + c_t = Q_t^*(s, a) - V_t^*(s)$$
$$= f_{\text{true}}(t, s, a) + \mathbb{E}_{S' \sim \mathcal{T}(\cdot | s, a)} \left[V_{t+1}^*(S')\right] - V_t^*(s)$$

where $c_t \in \mathbb{R}$ is a constant (which may depend on $t$, but not on $s$). This will yield $f_{\text{true}} = f_{\text{est}}$ provided $(t, s) \mapsto V_t^*(s)$ is a deterministic function of time (i.e. it is independent of $s$), and this is a necessary condition for nontrivial $\mathcal{T}$.

In other words, the identifiability issue discussed in the main body of this paper remains, as the demonstrator's trajectory distribution will depend on the state-action value function $Q_t^*$, rather than directly on the reward. Furthermore, this variation of guided cost learning generally corresponds to finding a reward which generates the observed policy, *and yields a value function $V^*$ which does not vary with the state of the system*. Of course, this reward will not usually be the same as that faced by the demonstrator, and so the results of guided cost learning are not guaranteed to generalize to agents with different transition probabilities.

We note that Balakrishnan et al. [2020] discuss the non-identifiability of costs in a MaxEntIRL approach. Their work focuses on building a projection under which rewards resulting in similar policies are mapped together, and then build a Bayesian estimation method for this projected data. What we have seen is that this approach is consistent (after the modifications discussed above), and will identify *some* cost function which gives the corresponding policy. For the entropy-regularized problem, our results precisely describe the kernel of this projection – it must correspond to different choices of the value function for the system.

## D   Appendix: Numerical examples of inverse reinforcement learning

In this section, we present a regularized MDP as in Section 3.1 to illustrate numerically the identifiability issue associated with inverse RL. In particular, we consider a state space $\mathcal{S}$ with 10 states and an action space $\mathcal{A}$ with 5 actions, with $\lambda = 1$. We compute optimal policies as in Section 2 and reconstruct the underlying rewards. As discussed in Section 3 the optimal policies and the transition kernel can be inferred from state-action trajectories, so will assumed known. We identify the state and action spaces with the basis vectors in $\mathbb{R}^{10}$ and $\mathbb{R}^5$ respectively, so can write $f(a, s) = a^\top R s$ for the reward function, and $\mathcal{T}(s'|s, a) = s^\top P_a(s')$ for the transition function. The true reward $R_{\text{tr}}$ and transition matrices $\{P_a\}_{a \in \mathcal{A}}$ are randomly generated and fixed; see Figures 1 and 2.

### D.1 Non-uniqueness of infinite-sample IRL

We first look at inverse RL starting from a single optimal policy $\pi_1$ with discount factor $\gamma_1 = 0.95$. We represent $\pi_1$ as a matrix $\Pi_1$ in $\mathbb{R}^{5 \times 10}$, where each column gives the probabilities of each action when in the corresponding state.

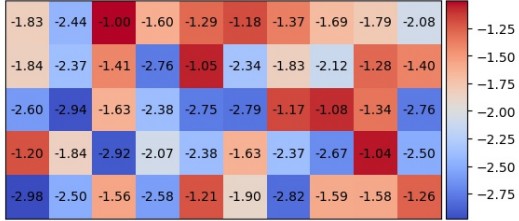

Figure 1: Underlying true reward matrix $R_{\mathrm{tr}}$

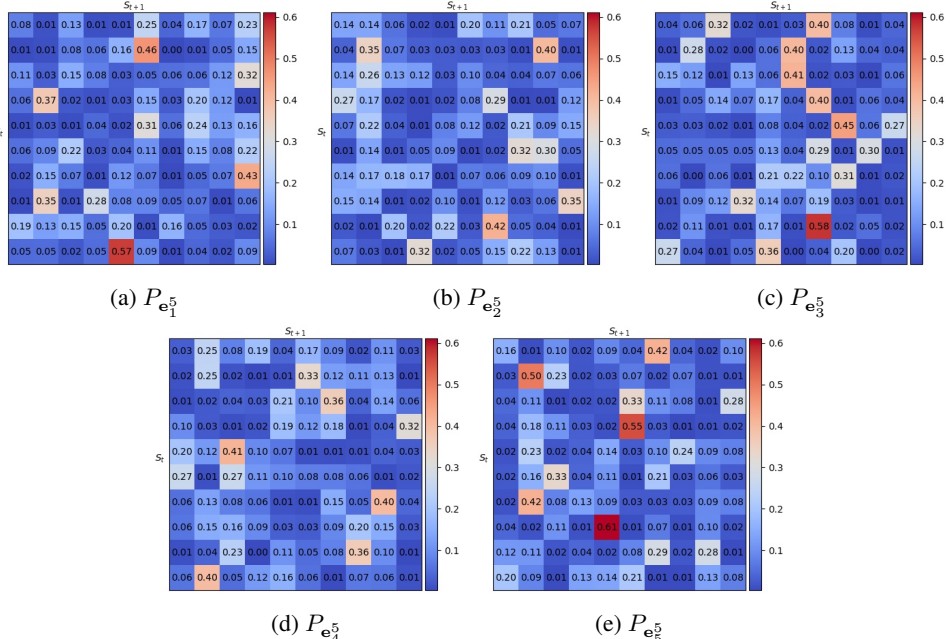

Figure 2: Underlying true transition kernel

To solve the inverse RL problem, we numerically find $R, v$ to minimize the loss

$$L_{\mathrm{sing}}(R,v) = \sum_{a \in \mathcal{A}} \sum_{s \in \mathcal{S}} \left[ a^\top \Pi_1 s - \exp\left\{ a^\top R s + \gamma_1 s^\top P_a v - v^\top s \right\} \right]^2.$$

An Adam optimizer is adopted with $\alpha = 0.002$, $(\beta_1, \beta_2) = (0.5, 0.9)$ with overall 2000 minimization steps. The experiments are conducted over 6 different random initializations, sampled from the same distribution as was used to construct the ground truth model. The training loss $L_{\mathrm{sing}}$ decays rapidly to close to 0, as shown in Figure 3a. This indicates that, after the minimization procedure comes to an end, the learnt reward matrix $\hat{R}$ reveals a corresponding optimal policy $\hat{\Pi}_1$ close to the true optimal policy $\Pi_1$; see also Figure 8 for a direct comparison.

However, when comparing the learnt reward $\hat{R}$ and the underlying reward $R_{\mathrm{tr}}$, as in Figures 4 and 5a, as well as the comparison between the corresponding value vectors as in Figure 5b, we can see that the true reward function $R_{\mathrm{tr}}$ has not been correctly inferred. Here this is not an issue of statistical error, as we assume full information on the optimal policy and the Markov transition kernel.

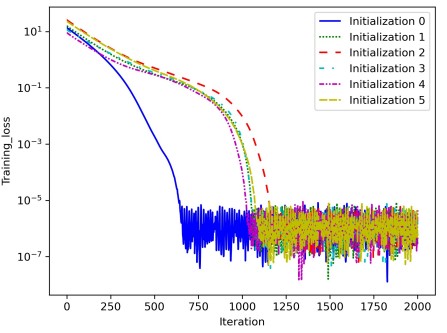 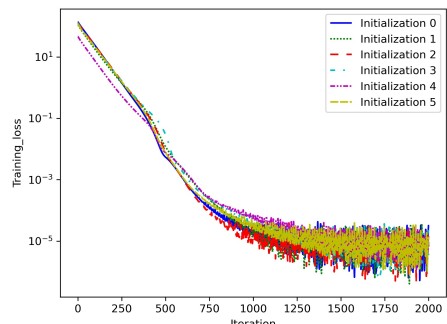

(a) Loss with one optimal policy, under $\gamma_1$    (b) Loss with two optimal policies, under $\gamma_1$ and $\gamma_2$

Figure 3: Training Losses

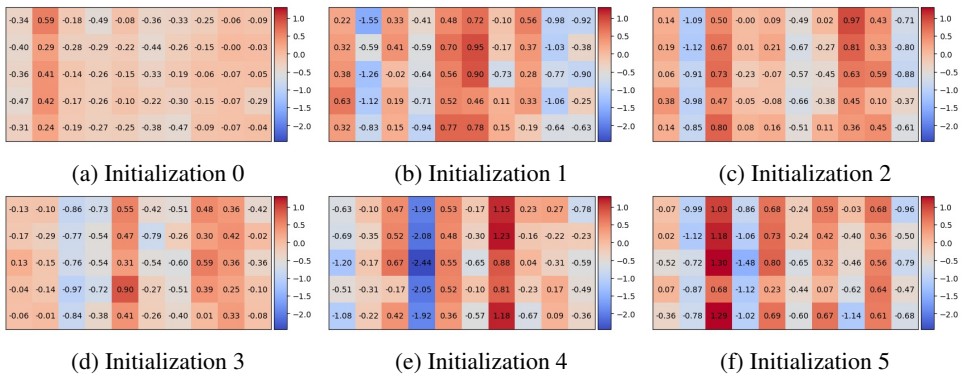

(a) Initialization 0    (b) Initialization 1    (c) Initialization 2

(d) Initialization 3    (e) Initialization 4    (f) Initialization 5

Figure 4: Learning from one optimal policy, under $\gamma_1$: difference $\hat{R} - R_{\mathrm{tr}}$ between learnt and true reward matrices

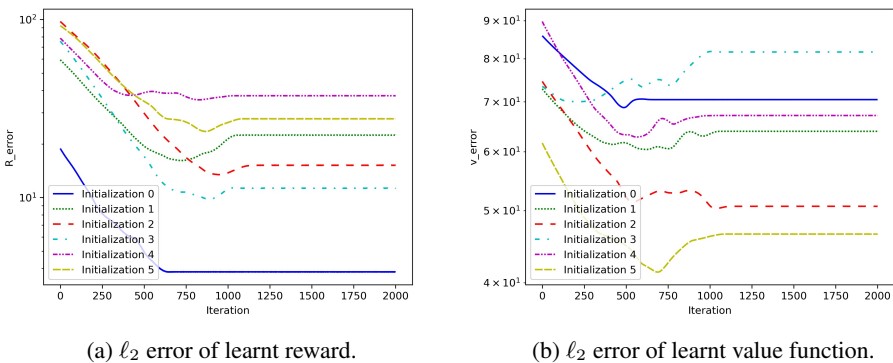

(a) $\ell_2$ error of learnt reward.    (b) $\ell_2$ error of learnt value function.

Figure 5: Learning from one optimal policy, under $\gamma_1$: comparisons

## D.2 Uniqueness of IRL with multiple discount rates

We now demonstrate that the issue of identifiability can be resolved if there is additional information on an optimal policy under the same reward matrix $R_{\mathrm{tr}}$ but different environment. Here, we assume we are given the policy $\Pi_1$ optimal with discount factor $\gamma_1 = 0.95$, and the policy $\Pi_2$ optimal with

discount factor $\gamma_2 = 0.25$. Correspondingly, the loss function for the minimization is adjusted to

$$L_{\mathrm{doub}}(R, v_1, v_2) = \frac{1}{2} \sum_{a \in \mathcal{A}} \sum_{s \in \mathcal{S}} \left[ a^\top \Pi_1 s - \exp\left\{ a^\top R s + \gamma_1 s^\top P_a v_1 - v_1^\top s \right\} \right]^2$$
$$+ \frac{1}{2} \sum_{a \in \mathcal{A}} \sum_{s \in \mathcal{S}} \left[ a^\top \Pi_2 s - \exp\left\{ a^\top R s + \gamma_2 s^\top P_a v_2 - v_2^\top s \right\} \right]^2.$$

An Adam optimizer is adopted with $\alpha = 0.005$, $(\beta_1, \beta_2) = (0.5, 0.9)$ with overall 2000 minimization steps. With the same set of 6 random initializations for the minimization procedure, the training loss $L_{\mathrm{doub}}$ also decays rapidly to close to 0. This again suggests that the learnt reward matrix $\tilde{R}$ can lead to policies $\tilde{\Pi}_1$ and $\tilde{\Pi}_2$, each optimal when using the corresponding discount factor $\gamma_1$ and $\gamma_2$, that are close to the given policies $\Pi_1$ and $\Pi_2$; see Figures 9 and 10. What differs from the single optimal policy case is that, with the additional information $\Pi_2$, we are able to consistently recover $R_{\mathrm{tr}}$ up to a constant shift; see Figures 6 and 7. Some numerical error remains, due to the optimization algorithm used, as seen by the fact the graphs in Figure 6 do still vary, and the error in the value function $v_1$ in 7(a). Nevertheless, the errors are an order of magnitude less than was observed in Figure 5 when using observations under a single discount rate.

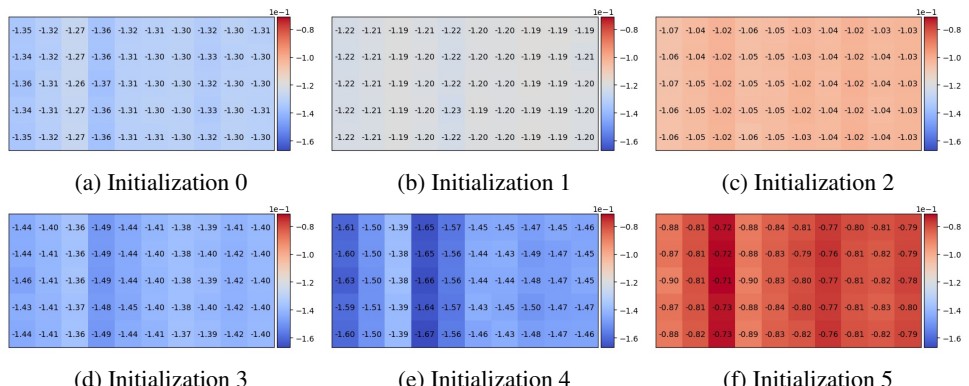

Figure 6: Learning from two optimal policies, under $\gamma_1$ and $\gamma_2$: difference $\tilde{R} - R_{\mathrm{tr}}$ between learnt and true $R$ matrices. Note scale of $10^{-1}$.

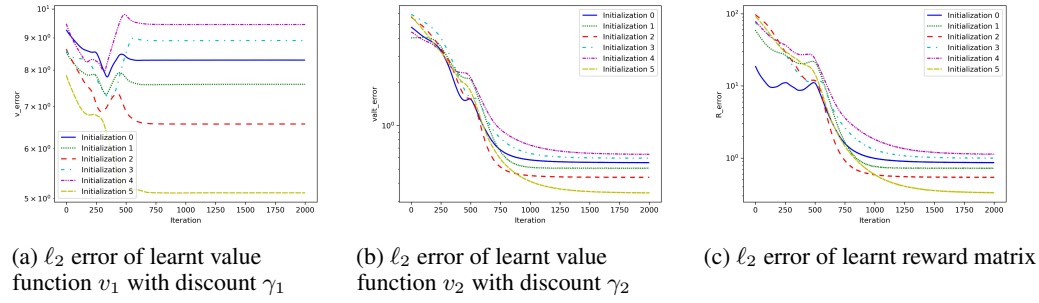

(a) $\ell_2$ error of learnt value function $v_1$ with discount $\gamma_1$

(b) $\ell_2$ error of learnt value function $v_2$ with discount $\gamma_2$

(c) $\ell_2$ error of learnt reward matrix

Figure 7: Two optimal policies under $\gamma_1$ and $\gamma_2$: comparisons

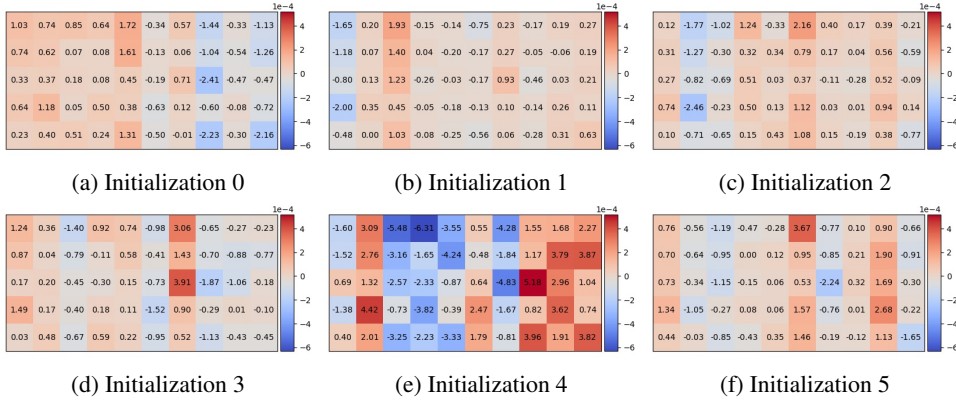

Figure 8: Learning from optimal policy under $\gamma_1$: difference $\hat{\Pi}_1 - \Pi_1$ between optimal policy under the learnt model and the true optimal policy. Note scale of $10^{-4}$.

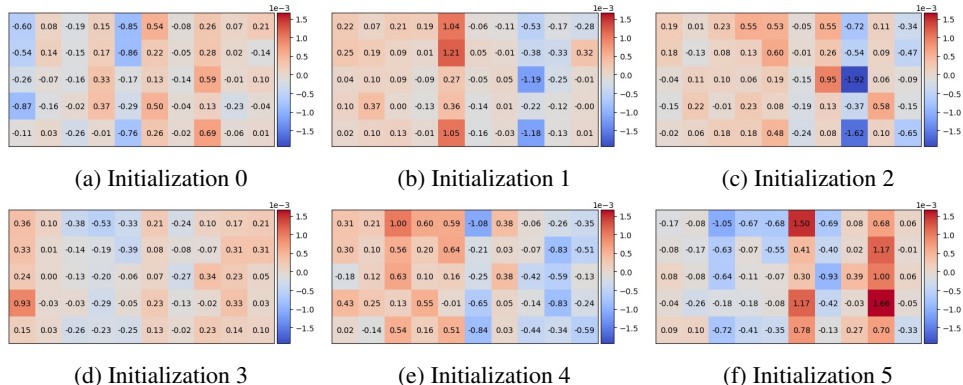

Figure 9: Learning from optimal policies under $\gamma_1$ and $\gamma_2$: difference $\tilde{\Pi}_1 - \Pi_1$ between learnt and true policies under $\gamma_1$. Note scale of $10^{-3}$.

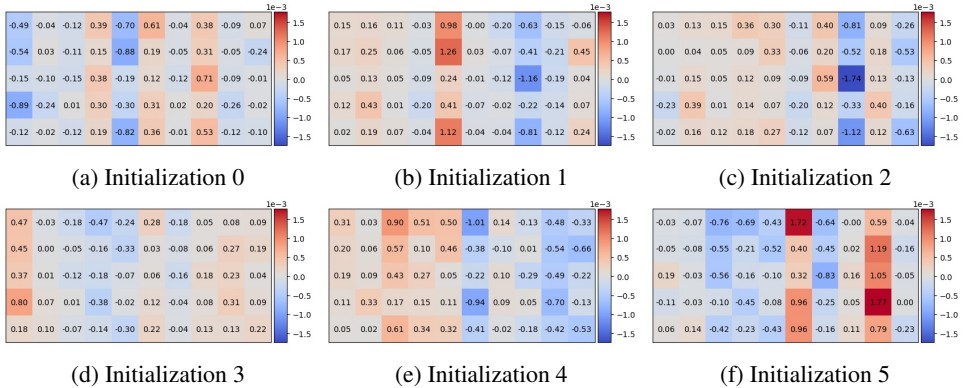

Figure 10: Learning from optimal policies under $\gamma_1$ and $\gamma_2$: differences $\tilde{\Pi}_2 - \Pi_2$ between learnt and true policies under $\gamma_2$. Note scale of $10^{-3}$.