# OpenReview forum: "Identifiability in inverse reinforcement learning"
_NeurIPS.cc/2021/Conference — NeurIPS 2021 Poster_

### Official Review · Reviewer_EpRH · 2021-07-15

**Rating:** 6
**Confidence:** 3

**Summary:**

This work presents theory and experiments on the problem of reward function non-identifiability in IRL. The key result is Theorem 1, which entails that the reward function (in the entropy-regularized MDP setting) cannot be uniquely identified given an unknown value function. Following this line of reasoning, the authors argue that it is possible to identify the reward function (up to a constant) provided we can disentangle immediate rewards from preferences over future states (e.g., with data from two agents with different discount factors in the LQG setting). This result is also demonstrated empirically using a small example.

**Limitations And Societal Impact:**

Yes, the limitations and societal impact are discussed.

**Main Review:**

This paper investigates a key problem in IRL, which is relevant for important issues, e.g., value alignment. The paper is generally well-written and the theoretical results are novel (to my knowledge) and elucidate why non-identifiability occurs, i.e., the unknown value function, and discusses when the true reward function can be retrieved. This theoretical study could potentially lead to novel downstream IRL methods. On the downside, the paper has issues that I hope the authors can address:
- The paper is not well-placed wrt the existing literature surrounding identifiability in IRL and the related works section misses out on closely-related prior work, e.g.,  (Kim et al, 2012) discusses conditions under which the reward function identifiable, and recent methods (e.g., Amin et al, 2017; Balakrishnan et al, 2020) have been proposed with non-identifiability in mind. See below for additional references.
- Can the authors provide some justification for the claim that the function class in Theorem 1 is unique? This is important as other reward function structures may also lead to the optimal policy, potentially giving rise to other forms of unidentifiability? Theorem 2 also relies on the uniqueness of the reward function. Also, can the authors clarify Remark 2? From my understanding, the results in Ng et al (1999) does not imply that non-identifiability solely arises from potential-based reward shaping functions.
- The experiments serve to support the claims, albeit in a very simple setting, and could be better presented. I suggest the authors reorganize the results; it would be helpful if all the key results were in the main paper rather than split off into the supplementary (e.g., fig 6/7 is in the supp). Fig 2 and 3 take up unnecessary space and can be summarized.

Overall, the work tackles an interesting problem, but due to the issues above, I remain uncertain and unconvinced about the significance of the work. I hope the authors can address my queries above.

References:
- “Reward Identification in Inverse Reinforcement Learning”, Kim et.al, ICML 2021
- "Repeated Inverse Reinforcement Learning”, Amin et.al., Neurips 2017
- “Towards Resolving Unidentifiability in Inverse Reinforcement Learning”, Amin et.al., 2016
- “Inverse Optimal Control with Linearly-Solvable MDPs”, Dvijotham et. al, ICML 2010
- “Efficient Exploration of Reward Functions in Inverse Reinforcement Learning via Bayesian Optimization”, Balakrishnan et.al,

### Post Response
Thank you for your response and the detailed comparison to other works. I agree that it was not reasonable to compare to Kim et al 2021 in the submitted version, but it would be good to include these comparisons in an updated version. While I think the contributions are of limited practical value at this time, there is some value to the findings and I have raised my score to a 6.

**Time Spent Reviewing:**

3

---

> ### Author Response · Authors · 2021-08-10
> **Response to Reviewer EpRH**
>
> We thank the reviewer for their comments and for acknowledging that the paper tackles important questions. As requested, we here provide a thorough comparison of our work to results from the literature pointed out by the reviewer; we would be happy to expand our literature review in a final version of the paper accordingly. The overall conclusion from the comparison is that our work helps unify different settings (within MaxEnt and entropy-regularized IRL) considered in the literature. Indeed our results are obtained without making significant simplifying structural assumptions (e.g. deterministic dynamics, a zero value function, state-only rewards, zero terminal rewards, etc.), are rigorously proven, and are sharp. Furthermore, from the way we set up the problem, many additional special cases can be easily derived. \
> We now address their criticism:
> * As discussed in our reply to reviewer TmNY, the paper Kim et al. (2021) only became publicly available after our submission; we refer to this response for more comments on the relationship between our work and theirs.
> * Comparison with Balakrishnan et al. (2020): Balakrishnan et al. acknowledge the issue of non identifiability as a statistical concern. Their paper uses a 'raw' maxEntIRL framework (as we discuss in Appendix C, and is discussed by Levine (2018), this does not correspond precisely to the entropy-regularized agent we consider, except in the deterministic setting with a constant value function). They develop a projection under which rewards resulting in similar policies are mapped together, and then build a Bayesian estimation method for this projected data. Our Theorem 1 in some sense describes the kernel of the projection in the entropy-regularized setting, in particular it shows that their method will generally not be able to reconstruct the 'true' reward function. This may not be a primary concern of theirs – their methods appear to consistently reproduce demonstrated behaviour for a given MDP. However, this will not result in optimal behavior if the reward is transferred to a different set of dynamics, and so understanding the limitations of their approach may be valuable.\
> Our approach also provides a natural version of their projection for the (dynamic, stochastic) entropy-regularized setting. As we have shown that the cost is only identified up to the choice of value function, we know that restricting our estimation to costs with a prescribed value function (e.g. $V(s)\equiv 0$) will precisely remove non-identifiability. This leads to a constrained optimization approach to estimation, which may be of use numerically. It also shows that if a consistent Bayesian method is used to estimate the rewards, the marginal prior and posterior laws on the value function will agree, no matter how many observations are made.
> * Comparison with Dvijotham and Todorov (2010): In this paper, rewards are assumed to be action-independent. Our formulation differ from theirs, due to the presence of discount factors, action-dependent rewards, and a more nuanced action space (they assume the transition probabilities can be controlled directly). In principle, this means that the results of Fu et al. (2017) apply (see also our response to reviewer YPRS for further discussion).
> * Comparison with Amin et al. (2017): Amin et al. (2017) consider an IRL problem with varying environment and action-independent rewards. They allow both for the case where the environment can be chosen, and the case where nature chooses an environment (adversarially). They suppose that each environment is treated separately by the demonstrator, which corresponds broadly to the setting of our Theorem 2. The key distinction is that our Theorem 2 demonstrates that given only two environments, satisfying the 'value-identifying' condition, it is possible ultimately to identify the reward fully. Amin et al. (2017) give related results, for example their Theorem 1, which shows that there exists a sequence of environments which quickly identify the true reward, or Theorem 4, in which the sequence of environments is chosen by nature. Their approach also adds the twist of allowing the IRL system to perturb the agents' rewards linearly, which is a key part of Theorem 4. However, this leads to a significant challenge, as the demonstrator agents are assumed to act on an infinite horizon, and many different environments may be needed in order to exploit their result.
> * Comparison to Amin and Singh (2016): Similar to the work of Amin et al. (2017), Amin and Singh (2016) studies an IRL problem with various environment and state-only reward functions. They show that, if the demonstrator is observed in multiple environments, the reward can be identified up to a scaling and shift (the scaling is natural, given they do not use an entropy regularization). They then consider how to select environments which yield the most information about the reward function. Again, our Theorem 2 addresses this question in slightly more generality (with action-dependent rewards, and an entropy regularization), and considers whether it is sufficient to observe behaviour only under two environments.
> * Our Theorem 1 provides the unique reward function $f$ such that both the entropy-regularized optimal policy $\pi^*$ is as observed, and the value function $V$ matches the (arbitrarily chosen) function $v$. No further structural assumptions are needed, and so we see that this completely characterizes the potential non-identifiability issue. The uniqueness is given simply by rearrangment of (6) – given the policy and value function, we can write $f$ explicitly and uniquely. The challenge in the proof is to show that $v$ can be specified arbitrarily and the system remains consistent; it is interesting that the proof of this fact requires no high-powered mathematical machinery – simply the explicit form of the value function (for a given reward) and a careful application of Jensen's inequality. This simplicity also ensures that the proof can be generalized (for example to continuous time and state-space problems) with very little variation.
> * The results in Ng et al. (1999) do not directly show non-identifiability for a given environment. As discussed in our response to reviewer YPRS, it is relatively straightforward to use these results to show that only a very particular form of reward (shaping potentials, where costs depend on future states) will lead to the same optimal policies under ***all*** transition probabilities. This implies that, if one had access to the optimal policy under all transition dynamics, the reward would be identifiable, up to a shaping potential. Given we restrict our attention to costs that do not depend on the future state, this corresponds to identification up to a constant. This essentially underlies the results of Amin et al. (2017), and is a weaker version of Theorem 2, where we only consider two sets of transition probabilities, and show that these are sufficient to identify the reward function fully.
> * We thank the reviewer for the comments regarding the organization of the paper, which was largely guided by the page limitations of the submission. We included Figures 2 and 3 in the main text as these illustrate our main point (the identifiability of the cost function), while figures 6 and 7 illustrate the convergence of the algorithm used; this seemed of secondary interest, as our results are independent of the numerical method selected.

---

### Official Review · Reviewer_YPRS · 2021-07-15

**Rating:** 5
**Confidence:** 4

**Summary:**

The paper considers the problem of reward identifiability in a maximum entropy inverse optimal control setting when access to the expert policy is available. It is shown that the reward function can only be recovered up to a state-dependent shaping term (which relates to the temporal credit assignment). If access to the optimal policies for two variants (different discount factor or different transition model) of a given MDP is available, it is shown that the reward function can be recovered (up to a constant).
The results are validated on numerical experiments on a finite MDP, with known transition model policies.

**Ethical Concerns:**

No ethical concerns.

**Limitations And Societal Impact:**

Sufficiently addressed.

**Main Review:**

Significance
----------------
Learning reward functions that generalize across different dynamics is one of the main open problems in inverse reinforcement learning. New insights under which assumptions such reward functions are recoverable could help to tackle this problem and would constitute an important contribution.

Originality
--------------
Unfortunately, I don't think that the paper presents any significant new insights. Fu et al. [2017] already noted that the log-policy corresponds to the soft advantage and thus a shaped reward function. They also noted that any reshaping with a potential function $\phi(s)$ results in the same optimal policy, and that this potential function corresponds to the value function of a reward function that can be computed as $r(s,a) = A(s,a) - \gamma E[\phi(s')] + \phi(s)$. Hence, I argue that its neither news that the expert policy alone only allows to recover the reward up to a temporal reshaping (relating to the Value function), nor is it news that the reward function can be recovered when in addition to the policy also the value function is known.

The paper also presents proof that the reward function can be recovered when having access to two optimal policies for different discount factors. While such proof seems to be novel, it is not surprising that the ill-posedness regarding the temporal credit assignment can be resolved by observing policies with different temporal weighting. Furthermore, I think that the derivations are straightforward and the insight little useful in practice.

Quality
-----------
The quality is good. The derivations seem correct, the numerical evaluation (or confirmation) make sense and shows the expected results.


Clarity
----------
The paper is well written. Eq.1 is missing parentheses around reward and entropy (discounting should also affect the entropy).


**Time Spent Reviewing:**

4

---

> ### Author Response · Authors · 2021-08-10
> **Response to Reviewer YPRS**
>
> We thank the reviewer for their comments, including the misplaced parenthesis in equation (1), for acknowledging that the paper is of good quality and the derivations and numerical results are correct.\
> The reviewer suggests that existing results in the literature cover our approach. We believe this is not the case. Our paper makes both novel contributions (for example the connection between the non-identifiability of rewards and the value function), and provides a simple approach for understanding identifiability issues in other situations, from a dynamic programming perspective. We also elucidate the connection between entropy regularization and the maxEnt approach to optimal control (Appendix C), proving a precise statement for the types of rewards which the maxEnt approach will construct.\
> We now address further specific points:
> * Comparison with Ng et al. (1999):  Ng et al. (1999) begin with a different structure to their cost function. In particular, their cost is of the form $f(S_t, A_t, S_{t+1})$. They then show that the addition of a shaping potential $\gamma \Upsilon(S_{t+1})- \Upsilon (S_t)$ leads to invariance of the optimal policies.\
> As pointed out by the reviewer, and in our Remark 2, by taking conditional expectations, for a fixed MDP this can be reduced to our setting of adapted costs (i.e. where the cost does not depend on the future state). However, this does not describe the space of all rewards corresponding to a given policy, for fixed transition dynamics. Using shaping potentials one can easily derive a sufficient criterion under which different rewards will lead to the same policy; in this sense Theorem 1 shows that this criterion is also necessary, with a simple and direct proof.\
> The results of Ng et al. (1999) in particular show that if costs are perturbed (other than with a shaping potential) there exists ***some*** set of dynamics such that the optimal policies would vary.  This is a weaker condition than our Theorem 2, where only a single alternative environment is needed to identify the reward function, and no structural assumptions (i.e. action-independence, decomposability of the system or deterministic dynamics) are required.
> * Comparison with Fu et al. (2017): As we mention this in our literature review section, using the results of Ng et al. (1999), as part of the proof of their Theorem 5.1/C.1, Fu et al. (2017) also show that there is at most one reward function (up to constants) which depends only on the current state, under their `decomposability' assumption on the MDP. Some aspects of their proof are somewhat opaque, in particular the role of the actions in the definition of decomposability. While we agree broadly with the approach of their proof, the definition allows counterexamples – for example, in the trivial situation with no actions and with all transitions possible, their result appears to claim that a state-dependent reward can be uniquely identified, despite no policy related information being available and all state-dependent rewards fitting equally well.\
> Our Theorem 1 can be easily extended to show that, assuming the transition dynamics depend on the actions in a sufficiently varied manner (which we believe is related to Fu et al.'s decomposability assumption), there exists at most one action-independent reward function (up to a constant) consistent with a given policy: All consistent rewards are represented by their value functions, so by taking the difference of two proposed action-independent costs, one obtains a linear system which can be constructed to identify the value function up to a constant. This proof (which yields a necessary and sufficient condition for identification) would be similar to Theorem 2, and we are happy to include it in the final version of our paper, if the reviewers feel this result for action-independent rewards is of interest.\
> Our Theorem 3 also demonstrates (but does not explicitly state) that there is at most one action-independent reward function (up to a constant) consistent with a given policy, in a linear-quadratic Gaussian setting; that is, where $a_{11}=a_{02}=a_{01}=0$. From this condition, an explicit (necessary and sufficient) criterion on the dynamics and observed policy under which a consistent action-independent reward exists can easily be derived: the slope of the dependence of the observed mean policy on the state needs to satisfy $k_1=\frac{\mu_s\mu_a+\sigma_s\sigma_a}{\mu_a^2 + \sigma_a^2}$. Again, we are happy to add a remark to this effect in the final version of the paper.
> * We feel that the clear identification in Theorem 1 of the manifold of costs (leading to a given policy) with the set of all value functions is a helpful conceptual contribution. The reviewer points out that the connection between the soft-advantage and the optimal policy is known (indeed, related results in convex duality are classical), however the one-to-one correspondence between rewards and the pair (optimal policy, value function) has not been widely explored.
> * We consider the role of discounting as an alternative resolution to non-identifiability. As the reviewer states, it is in some senses unsurprising that this yields a well-posed problem (as the fundamental issue of IRL is that the optimal control combines both the short-term rewards of interest and long-term rewards, as captured by the discounted value function), however this phenomenon has not been clearly stated in the literature.
> * While many of our derivations seem straightforward, this is a key strength of the paper and approach taken. In addition to leading to a readable text and quickly verifiable proofs, this means that many of our results can be easily extended: For example, Theorem 1 holds equally well, with the same proof, if the state space is infinite. By changing the transition probabilities to the infinitesimal generator (and similar minor changes of sums to integrals), the same proof holds in continuous time and state spaces, up to making appropriate continuity assumptions on the value function $v$. The paper mentions this fact at the beginning of Section 2, however the notational and technical overheads needed to give precise statements in continuous settings meant that we felt including these extensions was not beneficial to the main message of the paper.

---

> > ### Comment · Reviewer_YPRS · 2021-08-24
> > **Re: Response to Reviewer YPRS**
> >
> > Thank you for your reply.
> >
> > I agree that there is some novelty compared to Ng et al. (1999) since the MaxEnt setting is considered and the results are bit stronger by making it possible to identify the reward function from observing the expert in a second environment (e.g. discount factors).  Whether this contribution is sufficient to warrant acceptance is arguable. On the one hand, I was expecting that recovering the reward function when having observations from different observation would be possible but did not bother to formally proof it because I do not think that is very useful in practice, on the other hand formalizing the exact conditions may still turn out useful in the future, and there is little harm in publishing these results.
> >
> > Apart from this, I still do not see any other noteworthy novelty, e.g.
> > > however the one-to-one correspondence between rewards and the pair (optimal policy, value function) has not been widely explored.
> >
> > The one-to-one correspondence between rewards and policy/value function was very clear to begin with. The optimal policy is the log advantage function, and if you know the advantage and the value function, you also know the Q function, and in turn the reward function. Also the other direction (determinging policy and value function for a given reward function) is obviously possible.
> >
> > > Our paper makes both novel contributions (for example the connection between the non-identifiability of rewards and the value function) [...]
> > Also the connection to the value function is not novel, since the value function is literally a shaping potential (shaping a reward function with the value function generates the advantage function). So when you argue that reward+value function can not be identified, this seems to be equivalent to stating that reward+shaping can not be identified, just using a different term.

---

> > > ### Author Response · Authors · 2021-08-24
> > > **Re Re: Response to Reviewer YPRS**
> > >
> > > Thank you for the response.
> > >
> > > - We would like to point out that while using potential shaping one can easily derive a sufficient criterion under which different rewards will lead to the same policy; our Theorem 1 gives IF AND ONLY IF statement backed with direct proof.  We are not aware of any other results in the literature that also offers if and only if statements.
> > >
> > > - We believe that the identifiability of the reward given observations in two environments is interesting and critical for the application of IRL for policymaking, as we explained in our responses.  The key point result is new and hasn't been stated before. Furthermore,  as explained in response to EpRH, stronger assumptions are used in Amin et al (2016, 2017), who use all alternative environments (as does Ng et al 1999). This suggests having results involving only two environments are of some interest.

---

> > > > ### Comment · Reviewer_YPRS · 2021-09-01
> > > > **Re Re Re: Response to Reviewer YPRS**
> > > >
> > > > > We would like to point out that while using potential shaping one can easily derive a sufficient criterion under which different rewards will lead to the same policy; our Theorem 1 gives IF AND ONLY IF statement backed with direct proof. We are not aware of any other results in the literature that also offers if and only if statements.
> > > >
> > > > I agree and I referred to this when I said "[...]  the results are bit stronger by making it possible to identify the reward function from observing the expert in a second environment (e.g. discount factors)."
> > > >
> > > > > We believe that the identifiability of the reward given observations in two environments is interesting and critical for the application of IRL for policymaking, as we explained in our responses. The key point result is new and hasn't been stated before. Furthermore, as explained in response to EpRH, stronger assumptions are used in Amin et al (2016, 2017), who use all alternative environments (as does Ng et al 1999). This suggests having results involving only two environments are of some interest.
> > > >
> > > > I agree that there may exist potential applications, I just wanted to point out that for many cases the practicability is rather limited.
> > > >
> > > > I increased my rating to 5, because I underestimated the contribution in my initial review a bit (although I still think that it is rather small).

---

### Official Review · Reviewer_HJD7 · 2021-07-16

**Rating:** 6
**Confidence:** 4

**Summary:**

The authors consider the question of recovering the exact reward function from trajectories in inverse reinforcement learning. To this end, the authors consider the formalism of IRL with a maximum-entropy regularized objective which can be seen as a generalization of the MaxEntIRL objective. Using this view, the authors show that the degrees of freedom of the reward correspond to an arbitrary, desired value function. The authors show that the reward function can be fully recovered if the policy is known in two distinct MDPs with sufficiently different transition-dynamics and/or discount factors and give conditions for which this is the case. Similar theorems are given for finite MDPs as well as the linear quadratic regulator.


**Limitations And Societal Impact:**

-

**Main Review:**

I found the writing to be exceptionally clear and the proofs were easy to follow. One welcome change would be to clarify the contributions early on as they differ from what is typical in the IRL literature. Usually, identifiability is treated as a concern for stability and empirical analysis (of MaxEntIRL, for example) concentrates on the apprenticeship learning problem. In this paper, apprenticeship learning is not a concern as the authors explicitly assume that the policy can be fully recovered from demonstrated data. It is thus also unclear how well the algorithm would fare if expert data is limited and if there are scenarios where the insights from this paper could help recover optimal policies where behavioral cloning is insufficient.

While the given theorem is clear and, to the best of my knowledge, novel, it is also likely to be of limited use in direct applications. Not only do the authors assume access to the full expert policy, they furthermore assume access to the policy in two different MDPs with identical reward functions. Such MDPs can be constructed numerically, but are difficult to construct in practice.  In the background section, the authors claim that the transition dynamics can be recovered from expert trajectories; however, this is inconsistent with the proposed procedure where the dynamics have to be constructed. Furthermore, if the expert is a human demonstrator or a differently specified system, the discount factor is likely not known and should be treated as a modelling assumption. (In fact, human expert demonstrators are assumed to follow hyperbolic discounting, rather than exponential). Asking a demonstrator to provide demonstrations for two specific discount factors is therefore highly limiting. Discussion on how the proposed methods may generalize in the approximate case would be appreciated.

Numerical results are minimal and serve only as sanity checks. No application of the method is proposed. Overall, the contribution of this paper is clear, novel and well supported, but also limited.


**Time Spent Reviewing:**

3

---

> ### Author Response · Authors · 2021-08-10
> **Response to Reviewer HJD7**
>
> We thank the reviewer for their comments, and we're glad to hear that our efforts to make the paper readable have been appreciated. We accept that it would be helpful to further clarify the contributions of the paper in the introduction section, and will be happy to do so in the final version of the paper.\
> Our aim in this paper was not to focus on any specific numerical algorithm for the IRL problem, but rather to understand what is possible given perfect information. For this reason, our numerical examples were kept simple, to ensure that numerical difficulties did not obscure our results. Of course, in practice, the choice of approximation and inference procedure are critical, however focusing on these can obscure the structural problems which are independent of the algorithm used. By clearly presenting, and giving an interpretation to, the non-identifiability of the IRL problem, we hope to highlight these generic problems.\
> We now address further specific points:
> * Stability and empirical analysis: We completely agree that a natural follow up of this project would be to study the robustness and stability of our results e.g with respect to policy.  We studied an idealised case first to be able to show sharp results on identifiability (Theorem 1 and Theorem 2), in order to demonstrate what is possible even with perfect statistical information. Theorem 1 shows some forms of robustness already, in the sense that Theorem 1 specifies a reward function which is clearly continuous in its inputs (policies, value function, discount factor, transition dynamics); the robustness of Theorem 2 is also  easily seen to further depend on the well-posedness of the linear system in equation (9).\
> The explicit structure for the cost obtained in Theorem 1 may be of some use more generally: given knowledge of the value function, it demonstrates (unsurprisingly) that the logarithm of the action probabilities are the critical statistical quantities to estimate. Whether this is sufficient to lead to practical stability depends on the desired outcomes – we expect that highly sub-optimal rewards will correspond to small probabilities, and hence will be estimated with high statistical error (as the variance of the log probability explodes as the probability goes to zero, under most sampling regimes). It also allows us to identify the potential error due to estimation of the transition probabilities – if the value function is constant, then error in estimating transition probabilities leads to no error in the estimated costs. This is particularly of interest when we see the results in Appendix C (comparing with maxEnt IRL models on finite horizons), where we see that these tacitly assume that the value function is constant.\
> We believe the result of Theorem 1 may also be useful in settings where behavior cloning is insufficient (although this is not the main focus of this paper), as it demonstrates how the value function can be chosen arbitrarily, perhaps based on prior knowledge of the system, in order to infer natural rewards.
> *  Access to the policy in two different MDPs with identical reward functions:  We believe that assumption that one may have access to data sets with an agent optimising the same reward in two different conditions is natural in some settings. For example when applying IRL to economics, one typically has data on agents' choices before and after certain policy (e.g. changes to tax rules) have been implemented (i.e change to the environment). This is, in fact, the essence of Lucas' critique in economics – that agents policies will change when the environment is perturbed, leading to errors in predictions from classical macroeconomic models (where agents' actions are assumed to remain constant). This paper shows that it is, in principle (under strong rationality assumptions), possible for simple experimentation with the environment to reveal agents' preferences, and therefore to make valid predictions of their actions more generally.\
> The reviewer suggests that it is often implausible to ask agents to demonstrate behavior under multiple discount rates (and that agents may be more likely to use hyperbolic, rather than exponential discounting). We agree fully with this criticism, which strikes at the heart of the entire inverse RL project, independently of the approach taken. In many cases, assuming we can observe a large number of optimal decisions, from a consistent, rational agent, is a bold assumption. For some settings (e.g. in macroeconomic problems, when dealing with large banks and sovereign governments over short periods), rationality and consistency may be a reasonable approximation, but will generally be more tenuous when considering individuals.\
> More generally, working with alternative discounting rules (for example hyperbolic discounting) would involve modelling the demonstrator agents using a selection from the wide range of possible resolutions to time-inconsistent decision making (for example, naive/precommitment strategies, Strotz–Pollack-style game-theoretic approaches, forward utility constructions, compensator dual-optimality criteria, etc...). This would fundamentally change the nature of the optimizing agent, and would lead to a different inverse RL problem; given agents are inconsistent, their short-term costs also may not be of primary interest.\
> From this perspective, our primary result is a negative one (Theorem 1): Given only a single environment, even with perfect statistical estimation of all relevant quantities and classical rationality, it is still not possible to reconstruct agents' rewards from their actions (and we have characterized the precise form of non-identifiability in this setting).\
> Practically, our result does demonstrate (Theorem 2) that in this ideal world, identification is possible, given only observations under two environments. This is substantially simpler than is assumed, for example, in Fu et al. (2017), where the resolution to non-identification involves considering ***all*** possible sets of transition probabilities, or is restricted to action-independent rewards, or Amin et al. (2017), where it is possible to explore a range of environments (see comments to reviewer EpRH).\
> The reviewer also comments that, in this setting, the proposed estimation of the dynamics (in the background section) is invalid. We agree that this is a concern, but resolving it would depend closely on the type of perturbation made to the dynamics. Given sufficient observations in both regimes, it will be possible to estimate both sets of dynamics, however we expect this will lead to a significant robustness problem in practice, as the well-posedness of the linear system (9) will be difficult to establish.

---

### Official Review · Reviewer_TmNY · 2021-07-18

**Rating:** 4
**Confidence:** 3

**Summary:**

This paper studies the reward identifiability problem in inverse reinforcement learning, under a special family of MDPs -- entropy-regularized MDPs. The authors provide some mathematical deductions to show the identifiability is still an issue given entropy-regularization and side information from another policy with a different transition probability or a different discount factor but a consistent reward function may be sufficient to resolve it. They then construct a simple linear-quadratic Gaussian MDP to support their theorems and conduct numerical simulations. The empirical results validate their claims.

**Limitations And Societal Impact:**

The authors did not make extrapolative claims.

**Main Review:**

Originality: This paper makes some conceptual contributions in theorializing some intuitions of reward identifiability in IRL. I like the way the authors employ simple but appropriate mathematical language and examples to explain the idea. However, the idea they convey seems not novel to me, given the prior work from Ng et al. 1999. It is true that the entropy-regularization might be something not covered by Ng et al. 1999, but I wonder how the authors would compare this work with Kim et al., 2021, who study the same reward identifiability problem and deduce some structural properties for MDPs to obtain such identifiability.

Quality: The problem this paper studies is interesting -- to what degree the reward function is identifiable in IRL. The

Clarity: This paper is overall well-written. I like the way the authors build up everything step by step.

Significance: This paper may be interesting in terms of how the authors conceptualize the problem. But in my opinion, it is at least not as significant as Kim et al., 2021, who go beyond a trivial solution of introducing side information from another optimal policy. I would like to know my fellow reviewers' thoughts on this.

Kim et al. 2021, Reward Identification in Inverse Reinforcement Learning

**Time Spent Reviewing:**

3

---

> ### Author Response · Authors · 2021-08-10
> **Response to Reviewer TmNY**
>
> We thank the reviewer for their comments and stressing that the problem we study is interesting, the paper is very clearly written, and all the statements are correct.\
> Our work complements the now classic results of Ng et al. (1999), not only in that we incorporate the entropy regularization, but that we study the identifiability problem for a single MDP (or a pair of MDPs), with adapted costs (i.e. costs which do not depend on the future state). This allows a natural interpretation of the non-identifiability in terms of common quantities in control and RL: rather than an abstract shaping potential applied to future rewards, we show that non-identifiability is precisely described by a lack of knowledge of the value function. This approach reveals the structure of the IRL problem, for a single demonstrator, more clearly than has previously been done in the literature. Our results also focus on a general setting – we do not assume significant restrictions (for example, that the reward function is independent of the action chosen, or there is a fixed horizon with a zero terminal reward), which is commonly done in many previous works.\
> We now address further specific points:
> * Comparison to Ng et al.  (1999): We agree that the idea that the reward function, in general, cannot be identified is not new. However, Theorem 1, which shows that for a fixed regularised MDP, the cost function is parametrised by the value function, is new.  Furthermore, from Theorem 1 we see that many of the results in the IRL literature (implicitly) assume that value function is zero. This has a concrete interpretation: that many IRL methods make the tacit assumption that the demonstrator agent is myopic. Please also see Remark 2, and the response to reviewer YPRS, for further comparison with Ng et al. (1999)
> * Comparison to Kim et al. (2021): We were not aware of  Kim et al.'s work, which only became available (at least in the public domain) on the 21st of July (i.e. over a month after NeurIPS's submission deadline).  Fortunately, our and Kim's et al. works differ substantially and, in our view, complement each other.\
> The key difference is that Kim's et al. work studies deterministic MDP with maxEntRL on a finite time horizon and with zero terminal reward. In this setting, given appropriate structural assumptions, the reward is identifiable (see their Theorem 1). Our work is interested in identifiability in the general stochastic case, on infinite and finite time horizons (see Appendix C).  In our setting, we demonstrate that the space of rewards consistent with observed policies is parameterized by the value function, hence the best one can hope for is to identify the reward up to this class. Furthermore, we identify precise conditions that enable the identification of the reward when the environment can be varied.\
> Kim et al. in Section 5 provide further structural assumptions for identifiability. This is very interesting indeed, but again only holds in the deterministic case with zero terminal reward, and their main results focus on the case without discounting. In Appendix C we consider a more general finite time horizon problem with time inhomogeneous reward and non-trivial terminal cost and show that the results from the main body of our paper apply. This is because it is the zero terminal reward which plays a significant role in  Kim et al., as it provides knowledge of the (zero) value function at the terminal time. Combining this with a time homogeneity assumption is what enables the identifiability in this setting. Finally our results are independent of Kim et al.'s structural assumptions (they would hold equally if we had aperiodicity and strong recurrence of the Markov chains, which for us only is needed to justify the estimation of the dynamics and policy from a single trajectory). This explicitly limits how far one could hope to extend Kim et al.'s result.
> * We disagree with the reviewer that the solution we provided to demonstrate identifiability in the general case is trivial. For example, in economic problems, it may be possible to vary the system dynamics – this corresponds to the policy maker's problem in the Lucas critique, where agents' preferences are unknown, but government policy needs to be selected. Thanks to sharp results in Theorem 1 and 2, one can see that to be able to identify the reward, one needs to provide further information, and Theorem 2 provides a precise statement of one possible resolution, which does not depend on making significant assumptions about agents' preferences (e.g. that they are purely state dependent, or that there is a horizon with no terminal reward). This demonstrates that Lucas' critique can be overcome, as policy makers can (through experimentation, by perturbing discount rates or the transition probabilities) determine agents' preferences. In particular, we show that this is possible using a single alternative environment, and give a characterization of the condition which must be met. Our criterion does not make significant structural assumptions on the problem; we remain in the general time inhomogeneous setting with state and action dependent rewards. To the best of our knowledge, this result is not available in the existing literature. (See also our response to reviewer EpRH, where we compare with Amin et al. (2017), who make a stronger assumption in their analysis.)

---

### Decision · Program_Chairs · 2021-09-27

**Decision:**

Accept (Poster)

**Comment:**

Two reviewers recommended rejection of the paper (1x reject, 1x weak reject) and two reviewers recommended (weak) acceptance of the paper. Initially the reviewers raised concerns regarding novelty, significance, and relation to existing work. This was acknowledged by reviewers and some of them increased their score for the paper. I discounted the concern regarding comparison with Kim et al. as this paper was not available by the deadline of this conference (as also indicated by the authors). Overall the discussion of the authors with the reviewers and my own reading of the paper led me to the conclusion there are some interesting contributions in the paper and I am therefore recommending acceptance of the paper.
Nevertheless, the paper in its current form does not place the paper clearly in the context of existing results (as evidenced by a very short related work section and the initial confusion of the reviewers) and I strongly encourage the authors to make the relation to existing results more clear in final paper (e.g., by incorporating the responses they gave to reviewers during the discussion period).